# Calibrated Test-Time Guidance for Bayesian Inference

**Daniel Geyfman** [* 1]  **Felix Draxler** [* 1]  **Jan Groeneveld** [* 1]  **Hyunsoo Lee** [2]  **Theofanis Karaletsos** [3]  **Stephan Mandt** [1]

## Abstract

Test-time guidance is a widely used mechanism for steering pretrained diffusion models toward outcomes specified by a reward function. Existing approaches, however, focus on maximizing reward rather than sampling from the true Bayesian posterior, leading to miscalibrated inference. In this work, we show that common test-time guidance methods do not recover the correct posterior distribution and identify the structural approximations responsible for this failure. We then propose consistent alternative estimators that enable calibrated sampling from the Bayesian posterior. We significantly outperform previous methods on a set of Bayesian inference tasks, and set a new state-of-the-art PSNR in black hole image reconstruction. We publish our code at https://github.com/mandt-lab/Calibrated-Guidance.

## 1. Introduction

Diffusion models (Sohl-Dickstein et al., 2015; Song & Ermon, 2019; Ho et al., 2020; Song et al., 2021) and their flow matching variants (Liu et al., 2023; Lipman et al., 2023; Albergo & Vanden-Eijnden, 2023) have achieved tremendous success in sampling high-quality yet diverse images (Dhariwal & Nichol, 2021; Rombach et al., 2022) and videos (Yang et al., 2023; Ho et al., 2022) as well as scientific applications such as molecule generation (Hoogeboom et al., 2022) and protein-ligand docking (Corso et al., 2023).

A core benefit of diffusion models is their ability to sample from tilted distributions at test time: Take a pretrained model and guide the generation towards a desired outcome as specified by a potentially nonlinear reward function (Chung et al.,

---

[1]Department of Computer Science, University of California, Irvine, CA, USA [2]Seoul National University, Seoul, South Korea [3]Chan-Zuckerberg Initiative, Redwood City, CA, USA. Correspondence to: Felix Draxler <fdraxler@uci.edu>, Stephan Mandt <mandt@uci.edu>.

*Proceedings of the 43rd International Conference on Machine Learning*, Seoul, South Korea. PMLR 306, 2026. Copyright 2026 by the author(s).

2023). This enables applying pretrained diffusion models to inverse problems such as super-resolution, deblurring, denoising, style guidance, image editing, and others, without any additional training.

This approach is motivated by the goal of sampling from the Bayesian posterior, proportional to the diffusion model prior and a likelihood function that specifies the task (often called reward). In this context, we contribute:

- We find that **existing test-time guidance methods do not sample from the true Bayesian posterior** (Section 4), see Figure 1. We demonstrate that they use biased estimators for the diffused likelihood (Theorems 4.1 and 4.3 and Example 4.2), suggesting why they usually do not converge to the true posterior even with more compute.

- We propose Calibrated Bayesian Guidance (CBG), a new consistent yet tractable guidance framework for sampling the correct diffusion posterior in Section 5. Our framework supports non-differentiable objectives, and we propose practical approximations for fast sampling.

- Experimentally, our estimators accurately sample the Bayesian posterior on Bayesian inference and set a new state-of-the-art PSNR on a black-hole imaging task.

Together, we illustrate and address an important gap in the literature to accurately sample from Bayesian posteriors based on pretrained diffusion model priors.

## 2. Background

### 2.1. Diffusion Models

Diffusion models represent a data distribution $p(x)$ by iteratively transforming samples from a latent distribution $p(x_1)$ (usually a standard normal) through a stochastic or ordinary differential equation parameterized by a neural network $s_\theta(x_t, t)$ towards data samples of $p_\theta(x)$. The idea is to learn a neural network to reverse the addition of noise to the training data $x \sim p(x)$:

$$p(x_t \mid x) = \mathcal{N}(x_t; a_t x, b_t^2 I). \tag{1}$$

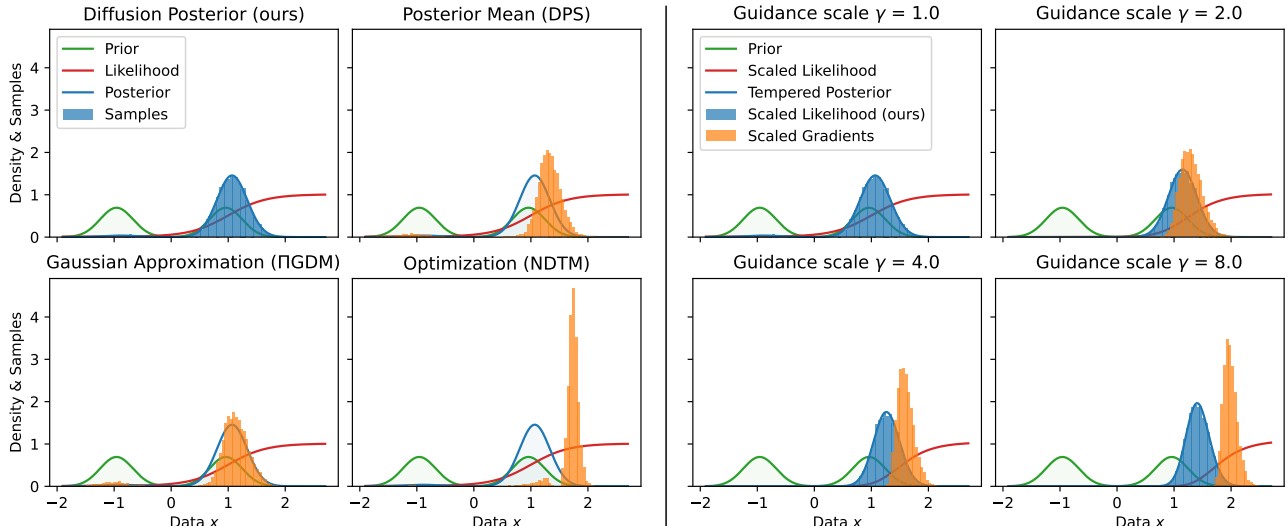

*Figure 1.* **We present a test-time guidance scheme to sample from calibrated Bayesian posteriors.** *(Left)* Our framework accurately samples the correct posterior (blue). Posterior mean (Equation (7)), posterior Gaussian (Equation (8)), and optimal control approximations (Equation (9)) to the diffused likelihood $p(y \mid x_t)$ yield uncalibrated samples (orange). *(Right)* Our framework can correctly sample from tempered posteriors $p(x \mid y, \gamma) \propto p(x)p(y \mid x)^{\gamma}$ (blue). Rescaling the noisy gradient by $\gamma$ leads to biased samples, even if the diffused likelihood $p(y \mid x_t)$ is accurately estimated (orange).

We adopt the convention that $t = 0$ corresponds to the data distribution (so $a_0 = 1$ and $b_0 = 0$), and $t = 1$ is the noisy latent ($a_1 = 0$ and $b_1 = 1$). We write $x = x_0$ for noise-free data samples.

The ordinary differential equation for sampling from a diffusion model is given by (Karras et al., 2022):

$$\frac{dx}{dt} = \frac{\dot{a}_t}{a_t}x - b_t^2 \dot{a}_t a_t \underbrace{\nabla_{x_t} \log p(x_t)}_{\text{learned } s_\theta(x_t, t)}. \tag{2}$$

Diffusion models are learned by minimizing the following loss:

$$\mathbb{E}_{t \sim \mathcal{U}[0,1], x \sim p(x), x_t \sim p(x_t|x)} \left[ \left\| \frac{a_t x - x_t}{b_t^2} - s_\theta(x_t, t) \right\|^2 \right]. \tag{3}$$

Other formulations of diffusion rewrite Equation (3) to instead predict the mean $f_\theta(x_t, t) \approx \mathbb{E}[x \mid x_t]$ or the instantaneous velocity $v_\theta(x_t, t) \approx \mathbb{E}_{x|x_t}[x - \epsilon \mid x_t]$, where $x_t = a_t x + b_t \epsilon$. They also introduce weighting functions $\lambda(t)$ that rescale the loss for better convergence, or sample $t \sim p(t)$. Similarly, different choices for the coefficients $a_t, b_t$ lead to different formulations of diffusion and flow matching.

For the experiments in this paper, we choose the schedules ourselves, or leverage pretrained models, in which case we adopt their diffusion schedule.

## 2.2. Bayesian Inference

We are interested in sampling from the Bayesian posterior:

$$p(x \mid y) \propto p(x)p(y \mid x). \tag{4}$$

Here, the prior $p(x)$ is usually given by training a diffusion model on data; in our experiments in Section 6.1, we find that it is even useful for analytic priors with tractable noising convolutions. The "observation" $y$ is an abstract placeholder for specifying the task: it could be a noisy measurement, but also covers arbitrary reward functions $r(x; y)$, potentially without an external signal $r(x; y) \equiv r(x)$, so that $p(y \mid x) = \exp(r(x; y))$.

Sampling from Equation (4) is a hard task in general. Diffusion models can be used here by learning them as a prior distribution $p(x)$. The core insight is that given a pretrained diffusion model, we can modify its prediction to correct from the diffusion prior to the Bayesian posterior via Bayes' rule:

$$\nabla_{x_t} \log p(x_t \mid y) = \underbrace{\nabla_{x_t} \log p(x_t)}_{\text{pretrained model}} + \underbrace{\nabla_{x_t} \log p(y \mid x_t)}_{\text{diffused likelihood}}. \tag{5}$$

While this modification is formally correct, the general form of the likelihood $p(y \mid x)$ in Equation (4) makes it impossible to analytically compute the diffused likelihood $p(y \mid x_t)$ on noisy data as required for Equation (5), even if we have a closed-form expression for $p(y \mid x)$.

Instead, we need to approximate this integral, which evalu-

ates to:

$$p(y \mid x_t) = \int p(x \mid x_t)p(y \mid x)dx, \qquad (6)$$

where $p(x \mid x_t)$ samples from the diffusion posterior.

The central goal of this paper is to identify valid approximations of the integral in Equation (6) that lead to sampling from the true Bayesian posterior $p(x \mid y)$ as defined in Equation (4).

### 2.3. Diffused Likelihood Approximation

Equation (6) is usually approximated to save costs. In this section, we show how these approximations prevent sampling from the correct posterior.

The common approximations in the literature are: The **Posterior Mean Approximation** computes the cost of the mean, instead of the mean of the cost (Chung et al., 2023):

$$p(y \mid x_t) \approx p(y \mid x = \mathbb{E}[x \mid x_t]). \qquad (7)$$

Intuitively, this is justified late in sampling when the diffusion posterior has small support relative to the curvature of the reward.

Evaluate Equation (6) with a **Posterior Gaussian Approximation** to the denoising distribution $p(x \mid x_t)$, as proposed as ΠGDM (Song et al., 2023a):

$$p(y \mid x_t) \approx \int p(y \mid x)\mathcal{N}(x; \mathbb{E}[x \mid x_t], \sigma_t^2 I)dx. \qquad (8)$$

The standard deviation is typically chosen as $\sigma_t^2 = b_t^2/(a_t^2 + b_t^2)$, the correct value if the prior $p(x)$ is a standard normal distribution.

**Optimal control** formulations such as nonlinear diffusion trajectory matching (NDTM) (Pandey et al., 2025b) steer the trajectory through a local optimization problem:

$$\tilde{x}_t = x_t + u_t, \text{where } u_t = \operatorname{argmin}_{u_t} \mathcal{L}(u_t, x_t) \qquad (9)$$

The loss function $\mathcal{L}(u_t, x_t)$ usually regularizes finite control $u_t$ plus maximizes the posterior mean approximation in Equation (7). The optimization over $u_t$ broadens the design space of the guidance framework. While this usually leads to higher rewards, it does not aim for calibrated posterior sampling.

We analyze the shortcomings of these estimators in Section 4 and propose a well-calibrated alternative in Section 5.

### 3. Related Work

Our training-free guidance framework is based on diffusion models. Diffusion models learn to generate data by iteratively denoising samples from a simple noise distribution, with foundational formulations ranging from discrete-time diffusion processes (Sohl-Dickstein et al., 2015; Ho et al., 2020) to continuous-time and score-based (Song & Ermon, 2019; Song et al., 2021) as well as velocity-based perspectives (Liu et al., 2023; Lipman et al., 2023; Albergo & Vanden-Eijnden, 2023). These methods have demonstrated strong empirical performance across modalities. Regarding the calibration of these models, existing work analyzes epistemic uncertainty (Jazbec et al., 2025), and improves tail behavior (Pandey et al., 2025a).

The test-time guidance framework we develop in this paper leverages pretrained diffusion models and adapts them to novel tasks. The first approaches to solving inverse problems using diffusion models consider *linear* tasks such as inpainting, super-resolution, and colorization (Kadkhodaie & Simoncelli, 2021; Song et al., 2021; Chung et al., 2022b; Kawar et al., 2022; Chung et al., 2022a). The methods strictly adhere to the task specification, not aiming to recover the Bayesian posterior.

Diffusion Posterior Sampling (DPS) (Chung et al., 2023) generalized this to solve nonlinear inverse problems using the posterior mean approximation. Several follow-up works extend DPS to more tasks, such as Universal Guidance for Diffusion Models (Bansal et al., 2024) and FreeDoM (Yu et al., 2023a). We find that the posterior mean approximation can yield high adherence to the reward, but the sampling does not follow the true Bayesian posterior diffusion.

To combat the inaccuracies in the posterior mean approximation, Pseudoinverse-Guided Diffusion Models (ΠGDM) (Song et al., 2023a) and C-ΠGDM (Pandey et al., 2024) approximate the diffusion posterior $p(x \mid x_t)$ by a normal distribution. This approximation is also used in Loss-Guided Diffusion (LGD) (Song et al., 2023b) as well as Diffusion Policy Gradient (DPG) (Tang et al., 2023). Again, we find this can be biased in general.

MPGD (He et al., 2024) projects the posterior mean to the data manifold and evaluate the likelihood there. This alleviates the problem that the reward function might be ill-defined off the data manifold, a common scenario for posterior approximations. However, this method is still limited by a single point estimate for the diffusion posterior.

Unified Training-Free Guidance (TFG) (Ye et al., 2024) unifies several design decisions underlying previous methods. Similarly, Nonlinear Diffusion Trajectory Matching (NDTM) (Pandey et al., 2025b) extends the design space, but ultimately both rely on the posterior mean approximation. Divide-and-Conquer Posterior Sampling (DCPS) (Janati et al., 2024) performs local Monte Carlo updates at every diffusion step.

In contrast to all these works, we formulate a consistent dif-

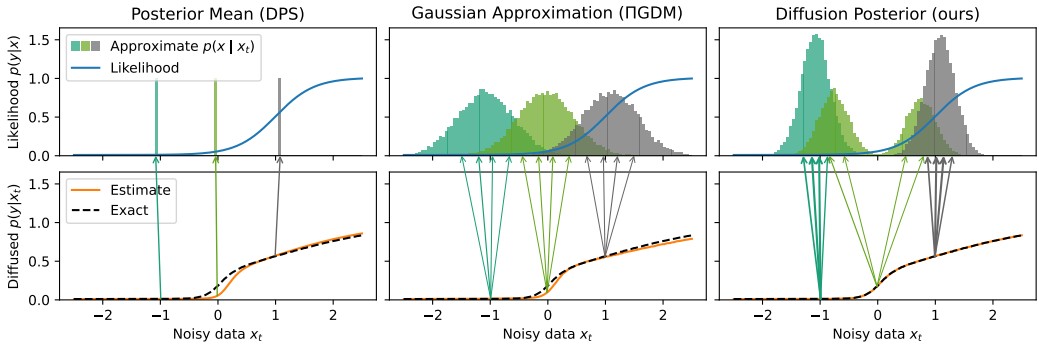

*Figure 2.* **Approximations of diffused likelihoods.** *(Left)* The posterior mean approximation in Equation (7) looks up the likelihood value at the mean of the diffusion posterior. *(Center)* Gaussian approximations to the posterior lead to inconsistent estimates that cannot be corrected by sampling more points. *(Right)* Our method relies on the true diffusion posterior $p(x \mid x_t)$, yielding arbitrary precision to determine the diffused likelihood $p(y \mid x_t)$ and its gradients.

fusion posterior sampling, where increased computational budget leads to calibrated samples. In concurrent work, Potaptchik et al. (2026) train few-step models to evaluate the same estimators as in Equations (16) and (20). While the focus of that work is on image synthesis, we focus on Bayesian inference. Also, by using the renoise trick, we can leverage pretrained one-step or few-step models.

A complementary body of work learns the diffused likelihood directly, which requires retraining for new tasks. This is achieved in classifier-free guidance (Ho & Salimans, 2021), which requires the likelihood to be representable as an input to the model. Other methods leverage reinforcement learning techniques to learn the diffused likelihood or finetune the teacher to a given reward (Li et al., 2024; Fan et al., 2023; Black et al., 2024; Uehara et al., 2024; Pandey et al., 2026).

## 4. Shortcomings of Existing Estimators

We now derive that the approximations of the diffused likelihoods in Section 2.3 yield uncalibrated Bayesian posteriors. In particular, we show that the approximations are always biased, apart from trivial cases.

### 4.1. Inconsistency of Diffused Likelihood Estimators

The estimators introduced in Equations (7) to (9) in Section 2.3 have one common pitfall: They are inconsistent estimators, meaning that even increasing the computational budget will not approximate the true diffused likelihood in Equation (6). Instead, they converge to a biased estimate.

First, we find that the posterior mean approximation in Equation (7) fails once the likelihood has a nontrivial maximizer on the support of the prior, which is common for example in the context of inverse problems:

**Theorem 4.1** (Informal). *Fix $0 < t < 1$. If the likelihood $p(y \mid x)$ has a nontrivial maximizer, then, unless the like-*

*lihood is constant in $x$, there exists some $x_t$ at which the posterior mean approximation in Equation (7) fails.*

This means that Equation (7) is not exact. The detailed statement is given in Appendix A.1, together with a mild regularity assumption. Intuitively, the proof shows that the maximum of the diffused likelihood is not the same as the maximum of the original likelihood: $\max_{x_t} p(y \mid x_t) \neq \max_x p(y \mid x)$, contradicting Equation (7).

Similarly, the Gaussian posterior approximation can be incorrect, and we give a concrete example below:

**Example 4.2.** *Let $p(x) = \frac{1}{2}(\mathcal{N}(x; \mu, 1) + \mathcal{N}(x; -\mu, 1))$ and $p(y \mid x) = \mathcal{N}(x; 0, \rho^2)$ for $\mu, \rho^2 > 0$. Then, for every $t$, there exists $x_t \in \mathbb{R}$ such that Equation (8) does not hold.*

Please see Appendix A.2 for the detailed derivation.

Figure 1 (left) shows how this leads to methods relying on these estimates to sample from incorrect posteriors: Every step in the guided diffusion process in Equation (5) follows an approximation of the guidance term that ultimately leads to bad calibration. Figure 2 visualizes how the different approaches approximate the diffused likelihoods.

In Section 5, we will derive novel estimators for the diffused likelihoods that are consistent: Increasing the computational budget for the diffused likelihood makes the estimator converge to the true value.

### 4.2. Bias of Guidance Scales

However, even if we have access to a correct $p(y \mid x_t)$, there is another common bias introduced when guiding diffusion models: It occurs when one tries to vary the importance of the prior $p(x)$ relative to the likelihood $p(y \mid x)$ using a parameter $\gamma$:

$$p(x \mid y, \gamma) \stackrel{\text{def}}{\propto} p(x)p(y \mid x)^{\gamma}. \qquad (10)$$

This is referred to as a tempered likelihood posterior in Bayesian inference (Wuttke et al., 2014; Mandt et al., 2016; Friel & Pettitt, 2008; Pawlowski et al., 2017).

In the context of diffusion models, $\gamma$ is referred to as the classifier or the guidance scale (Ho & Salimans, 2021; Dhariwal & Nichol, 2021). It is usually implemented by rescaling the likelihood term in Equation (5):

$$\nabla_{x_t} \log p(x_t) + \gamma \nabla_{x_t} \log p(y \mid x_t). \tag{11}$$

Assume now that we have access to the ground truth $p(y \mid x_t)$ for $\gamma = 1$. Equation (11) would imply that the *tempered* diffused likelihood reads:

$$p(y \mid x_t, \gamma) \overset{?}{=} p(y \mid x_t)^\gamma. \tag{12}$$

However, this is not correct. According to the definition in Equation (6), the guidance scale needs to be applied *within the integral*:

$$p(y \mid x_t, \gamma) \propto \int p(x \mid x_t) p(y \mid x)^\gamma dx. \tag{13}$$

In fact, we can show that Equation (12) is only correct in the trivial case where the likelihood is constant, in which case guidance has no effect:

**Theorem 4.3** (Informal). *Fix $0 < t < 1$ and $\gamma \in \mathbb{R}_+ \setminus \{0, 1\}$. Unless $p(y \mid x)$ is constant, there exists some $x_t$ at which the naive tempered diffused likelihood Equation (12) fails.*

The detailed statement and proof are given in Appendix A.3. A similar result by Sun et al. (2024) previously showed that the diffusion trajectory under Equation (12) is inconsistent with the true diffusion forward process.

So whenever the likelihood is nontrivial, picking Equation (12) instead of Equation (13) results in an error. Figure 1 (right) shows how the incorrect Equation (11) as opposed to the correct Equation (13) leads to sampling from an incorrect tempered posterior.

Notably, this restriction not only applies to existing test-time guidance methods, but it also applies to classifier-free guidance (Ho & Salimans, 2021), for which we can only extract the *untempered* diffused likelihood gradient:

$$\nabla_{x_t} \log p(y \mid x_t) \approx \nabla_{x_t} \log p_\theta(x_t \mid y) - \nabla_{x_t} \log p_\theta(x_t). \tag{14}$$

Only a tempered conditional model $p_\theta(x_t \mid y, \gamma)$ can extract the correct tempered diffused likelihood gradient.

This means rescaling the diffused likelihood gradient is not enough for calibrated tempered inference.

Together, these results imply that no amount of additional computation can correct the bias in existing methods: they converge to the wrong distribution. We now derive a consistent test-time guidance framework that is immune to the shortcomings in Sections 4.1 and 4.2.

## 5. Calibrated Bayesian Guidance (CBG)

In the light of the shortcomings of existing estimators in Section 4, we now present a novel guidance framework that enables consistent sampling from the true Bayesian posterior, both with and without tempering the likelihood. The central idea is to directly approximate the integral in Equation (6).

### 5.1. Differentiable Rewards

Assuming $p(y \mid x)$ is differentiable everywhere, we can compute the gradient of the diffused likelihood using the reparameterization trick:

$$\nabla_{x_t} \log p(y \mid x_t)$$
$$= \frac{1}{p(y \mid x_t)} \nabla_{x_t} \int p(x \mid x_t) p(y \mid x) dx$$
$$= \frac{1}{p(y \mid x_t)} \int \nabla_{x_t} p(y \mid x = g_t(x_t; \epsilon)) p(\epsilon) d\epsilon. \tag{15}$$

Here, $g_t(x_t; \epsilon)$ samples from $p(x \mid x_t)$ such that it is differentiable with respect to $x_t$, and $\epsilon$ is the randomness that is used to sample from it.

This makes the gradient of the diffused likelihood straightforward to estimate:

$$\nabla_{x_t} \log p(y \mid x_t)$$
$$\approx \frac{1}{\sum_i p(y \mid x^{(i)})} \sum_{i=1}^{K} \nabla_{x_t} p(y \mid x = g_t(x_t; \epsilon^{(i)})). \tag{16}$$

Here, $\epsilon^{(i)} \sim p(\epsilon)$ samples the randomness in the sampler $g$ for samples $i = 1, \ldots, K$. While naive diffusion sampling such as DDPM (Ho et al., 2020) can be used to sample from $x \sim p(x \mid x_t)$, we use few- or one-step models as detailed in Section 5.3. We call Equation (16) **Gradient-Based Calibrated Bayesian Guidance**.

In practice, we work with log-likelihoods $\log p(y \mid x)$ and average using $\mathrm{softmax}(w_i)$ for numeric stability.

Equation (16) yields a consistent sampling procedure: as $K \to \infty$, any bias vanishes. This stands in contrast to the estimators considered in Section 2.3, which remain biased even asymptotically. While consistency requires averaging over multiple samples to obtain a reliable guidance signal, the additional computation can be viewed as the necessary cost of eliminating systematic bias rather than a fundamental limitation of the estimator itself.

## 5.2. Non-Differentiable Rewards

The reparameterization estimator Equation (16) has the major drawback that it is not compatible when computing gradients through the sampling process $p(x \mid x_t)$ and/or the likelihood $p(y \mid x)$ is computationally expensive or intractable.

To this end, we propose a REINFORCE estimator (Williams, 1992) to compute the gradient of the diffused likelihood in Equation (5):

$$\nabla_{x_t} \log p(y \mid x_t)$$
$$= \frac{1}{p(y \mid x_t)} \nabla_{x_t} \int p(x \mid x_t) p(y \mid x) dx \qquad (17)$$
$$= \frac{1}{p(y \mid x_t)} \int p(x \mid x_t) p(y \mid x) \nabla_{x_t} \log p(x \mid x_t) dx.$$

Inserting into Equation (5), we absorb $\nabla_{x_t} \log p(x_t)$ into the integral and find for the diffused posterior score:

$$\nabla_{x_t} \log p(x_t \mid y) \qquad (18)$$
$$= \frac{1}{p(y \mid x_t)} \int p(x \mid x_t) p(y \mid x) \nabla_{x_t} \log p(x_t \mid x) dx.$$

By Equation (1), we can replace

$$\nabla_{x_t} \log p(x_t \mid x) = \frac{a_t x - x_t}{b_t^2}. \qquad (19)$$

Evaluated using a finite set of samples $x^{(i)} \sim p(x \mid x_t)$ for $i = 1, \ldots, K$, we find our estimator:

$$\nabla_{x_t} \log p(x_t \mid y) \approx \frac{1}{\sum_i w_i} \sum_{i=1}^{K} w_i \frac{a_t x^{(i)} - x_t}{b_t^2}. \qquad (20)$$

The likelihood enters as weights $w_i = p(y \mid x^{(i)})$. We call Equation (20) **Gradient-Free Calibrated Bayesian Guidance**.

Equation (20) is again a consistent estimator, meaning that it reduces bias with more samples. Again, the estimator relies on samples $x^{(i)} \sim p(x \mid x_t)$. In the next section, we discuss how to obtain these samples efficiently.

## 5.3. Sampling from the Diffusion Posterior $p(x \mid x_t)$

Both estimators in Equations (16) and (20) rely on samples from the diffusion posterior $x \sim p(x \mid x_t)$. The straightforward way to obtain them is stochastic diffusion sampling such as DDPM (Ho et al., 2020). However, this scales poorly: drawing $K$ samples from $x_t$ down to $x$ with an $N$-step sampler, across an outer denoising loop of $N$ noise levels, requires $KN(N-1)/2 \in \mathcal{O}(KN^2)$ model evaluations. This can quickly run into computational bottlenecks, especially for large $K$ and large models.

One mitigation is to limit the number of DDPM iterations for the per-step sample generation to a maximum of $M$. This reduces the quadratic complexity to a linear dependence $O(MNK)$ on the total number of diffusion steps. Empirically, we found that this yields better quality for the same compute than reducing $K$ instead.

To reduce compute further, we propose to either use an analytic expression for $p(x \mid x_t)$ if available (such as in the Bayesian inference tasks in Section 6.1), or to use a pretrained one-step or few-step model (Song et al., 2023c; Draxler et al., 2024; Geng et al., 2025; Frans et al., 2025) to sample from it. These approaches reduce the cost to $\mathcal{O}(KN)$.

In detail, we propose to use *deterministic* one-step models $x = f_t(x_t)$ to sample from the stochastic posterior $p(x \mid x_t)$ via renoising:

$$x^{(i)} = f_t\Big(a_t f_t(x_t) + b_t \epsilon^{(i)}\Big), \quad \epsilon^{(i)} \sim \mathcal{N}(0, I). \quad (21)$$

Here, $a_t, b_t$ are the schedule coefficients from Equation (1), and $\epsilon^{(i)}$ is independent noise. Equation (21) applies $f_t$ to obtain a deterministic prediction $\hat{x} = f_t(x_t)$, perturbs it back to noise level $t$ with independent noise $\epsilon^{(i)}$, and denoises once more. The renoising step itself has been used previously in other contexts (Wang et al., 2023; Yu et al., 2023b; Lugmayr et al., 2022). Even though this construction is heuristic, we empirically observe identical performance to using inner diffusion rollouts at orders of magnitude faster inference (e.g. Table 2). Concurrent to this work, Potaptchik et al. (2026) propose to *learn* a stochastic one-step sampler $x = f_t(x_t, \epsilon)$ for $p(x \mid x_t)$ instead. However, at the time of writing, these models are not readily available for many modalities, while deterministic one-step models often are.

## 5.4. Comparing Gradient-Free and Gradient-Based Methods

One might think that being based on REINFORCE, the gradient-free estimator in Equation (20) has larger variance than the gradient-based estimator in Equation (16), as discussed in (Paisley et al., 2012; Ruiz et al., 2016; Miller et al., 2017).

Interestingly, this general rule does not apply to our estimators, since they involve a self-normalization term of dividing by the sum of the weights. Empirically, we find that the variance of the gradient-free estimator in Equation (20) is in general lower than that of the gradient-based estimator in Equation (16). See Section C for a concrete example.

The reason is that the two estimators behave quite differently in the presence of sharp likelihoods: The gradient-free estimator reduces to a weighted sum, as can be seen by

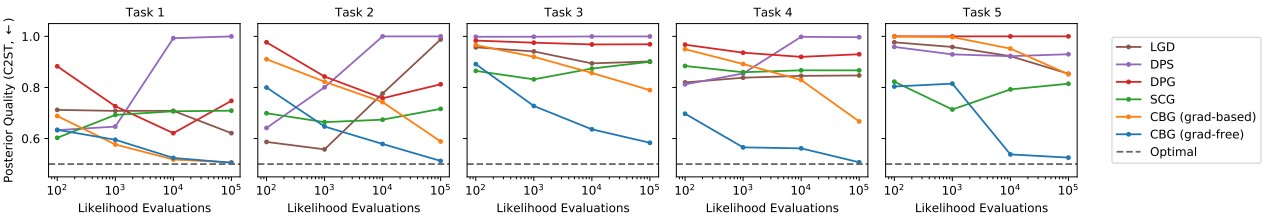

*Figure 3.* Empirical performance on Bayesian Inference tasks. Performance is measured in C2ST (lower is better, ↓) (Friedman, 2004), comparing the distribution of guided samples to those of ground truth samples. Our methods improve performance with more compute, while other test-time adaptation methods are limited due to their approximations to diffused likelihood gradients.

*Table 1.* Best method performance on Bayesian Inference C2ST (↓), standard deviations in parentheses.

| Algorithm | Task 1 | Task 2 | Task 3 | Task 4 | Task 5 | Average |
|---|---|---|---|---|---|---|
| *Diffusion Methods* | | | | | | |
| CBG (gradient-free, ours) | 0.505 (0.004) | 0.513 (0.009) | 0.584 (0.063) | **0.507 (0.006)** | **0.525 (0.028)** | **0.527** |
| CBG (gradient-based, ours) | 0.507 (0.005) | 0.589 (0.048) | 0.789 (0.089) | 0.667 (0.019) | 0.852 (0.038) | 0.681 |
| DPS (Chung et al., 2023) | 0.633 (0.024) | 0.641 (0.032) | 0.998 (0.002) | 0.812 (0.023) | 0.929 (0.045) | 0.803 |
| LGD (Song et al., 2023b) | 0.621 (0.049) | 0.558 (0.013) | 0.894 (0.042) | 0.820 (0.028) | 0.854 (0.117) | 0.750 |
| DPG (Tang et al., 2023) | 0.621 (0.078) | 0.758 (0.027) | 0.968 (0.015) | 0.920 (0.023) | 1.000 (0.000) | 0.872 |
| SCG (Huang et al., 2024) | 0.603 (0.037) | 0.664 (0.028) | 0.832 (0.027) | 0.860 (0.017) | 0.714 (0.131) | 0.735 |
| Langevin+Noisy Classifier (Song & Ermon, 2019) | 0.506 (0.002) | 0.520 (0.005) | 0.633 (0.017) | 0.642 (0.083) | 0.538 (0.030) | 0.568 |
| Langevin+REINFORCE (Song & Ermon, 2019) | **0.502 (0.003)** | 0.514 (0.004) | 0.604 (0.009) | 0.705 (0.015) | 0.540 (0.013) | 0.573 |
| MCMC No-U-Turn Sampler (Hoffman & Gelman, 2014) | 0.514 (0.005) | 0.511 (0.004) | 0.627 (0.005) | 0.512 (0.007) | 0.603 (0.046) | 0.553 |
| *Likelihood-free Methods* | | | | | | |
| NLE (Papamakarios et al., 2019) | 0.515 (0.009) | **0.506 (0.004)** | 0.699 (0.069) | 0.731 (0.023) | 0.668 (0.094) | 0.624 |
| NPE (Papamakarios & Murray, 2016) | 0.506 (0.004) | 0.509 (0.005) | 0.831 (0.052) | 0.555 (0.015) | 0.542 (0.024) | 0.589 |
| NRE (Hermans et al., 2020) | 0.536 (0.031) | 0.631 (0.034) | 0.919 (0.039) | 0.734 (0.019) | 0.629 (0.056) | 0.690 |
| REJ-ABC (Pritchard et al., 1999) | 0.802 (0.023) | 0.909 (0.028) | 0.961 (0.020) | 0.772 (0.034) | 0.664 (0.040) | 0.822 |
| SMC-ABC (Beaumont et al., 2009) | 0.726 (0.032) | 0.794 (0.041) | 0.963 (0.018) | 0.664 (0.011) | 0.663 (0.048) | 0.762 |
| SNLE (Papamakarios et al., 2019) | 0.519 (0.008) | 0.509 (0.005) | **0.578 (0.029)** | 0.624 (0.089) | 0.571 (0.050) | 0.560 |
| SNPE (Greenberg et al., 2019) | 0.507 (0.004) | 0.507 (0.005) | 0.666 (0.061) | 0.533 (0.014) | 0.530 (0.026) | 0.545 |
| SNRE (Hermans et al., 2020) | 0.515 (0.004) | 0.536 (0.007) | 0.721 (0.058) | 0.542 (0.016) | 0.563 (0.032) | 0.575 |

extracting Tweedie's estimate from Equation (20):

$$\mathbb{E}[x \mid x_t, y] \approx \sum_i \frac{w_i}{\sum_j w_j} x^{(i)}. \qquad (22)$$

As the weights $w_i$ are typically of different orders of magnitude in practice, it often considers only the subset of samples with relatively high likelihood. The diffusion process will then move towards these samples in a region of high likelihood, discarding the others.

For the gradient-based estimator in Equation (16), the estimate is dominated by gradients associated with the highest-likelihood samples. These gradients can still be large, since small gradients occur only in a narrow neighborhood around the likelihood maximizer. Moreover, they are effectively amplified because the normalization term is determined by weights that are smaller than the (unobserved) maximum likelihood value:

$$\nabla_{x_t} \log p(y \mid x_t) \approx \sum_i \frac{\nabla_{x_t} w_i}{\sum_j w_j} \qquad (23)$$

As a consequence, a large gradient can induce a disproportionately large update. Reducing this effect requires a larger number of samples, so that local gradients average out into

a stable global direction and the maximum observed weight becomes a less noisy estimate of the true tail behavior.

From a practical perspective, the reparameterization estimator also requires computing gradients through the diffusion sampling, which increases computational cost and increases memory consumption. We therefore do not consider it for high-dimensional experiments.

## 6. Experiments

We now confirm the quality of our framework for practical Bayesian inference tasks. We find that CBG outperforms other test-time guidance methods on tasks where the prior and the likelihood are given as closed-form expressions, as well as on a scientific inverse problem where the prior is given by a pretrained diffusion model of black hole images.

### 6.1. Bayesian Inference Benchmark

In this experiment, we evaluate a diverse collection of diffusion guidance methods on a benchmark of Bayesian inverse problems proposed by Lueckmann et al. (2021). This benchmark is designed to test methods on their fit to the true Bayesian posterior $p(x \mid y)$ at varying computational bud-

gets. To this end, the benchmark provides reference samples from the true posterior.

For the diffusion posterior $p(x \mid x_t)$ in Equations (16) and (20) as well as for the posterior mean $\mathbb{E}[x \mid x_t]$ in Equations (7) and (8), we leverage closed-form expressions from the true prior $p(x)$ of these tasks (see Section F). This is possible because the prior distributions are analytic expressions such as normal and uniform distributions. This makes the experiments blazingly fast: A single inference takes on the order of milliseconds.

Figure 3 shows how both our gradient-free and gradient-based estimators improve their C2ST towards the optimal value of 0.5 as the compute budget is increased. Other test-time guidance methods generally do not improve performance with more compute.

Additionally, Table 1 shows the best classifier score for each method over all hyperparameters. The gradient-free method achieves the best distributional fit for every task by far compared to all other test-time guidance methods. We also evaluate against a collection of likelihood-free methods that rely only on samples $y^{(i)} \mid x$ and a target $y$, instead of an exact likelihood $p(y \mid x)$. Our CBG method also outperforms these likelihood-free methods on average.

Together, we see that our estimators perform well on Bayesian inverse tasks, and outperform a significant collection of likelihood-based and likelihood-free methods.

## 6.2. Black Hole Imaging

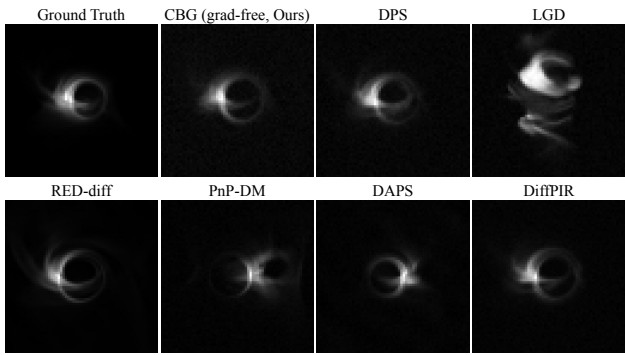

*Figure 4.* Uncurated comparison of CBG+DDPM with other test-time guidance methods on the black-hole imaging task proposed by Lueckmann et al. (2021). Despite computing no likelihood gradient, CBG is able to reconstruct the ground truth samples well.

In this experiment, we apply our method to scientific inference, where having a calibrated posterior is important. In particular, we choose the black hole imaging task proposed by Zheng et al. (2025) with a diffusion prior trained on images by Mizuno (2022). The task provides 100 problem instances, each consisting of a reference solution $x^*$ as well as a corresponding radio-telescope measurement $y$. The likelihood function $p(y \mid x)$ measures the compatibility

*Table 2.* Peak signal-to-noise ratio (PSNR, higher is better) on 100 black hole imaging tasks. Standard deviations over tasks in parentheses. Per-sample inference time measured on a single NVIDIA TITAN RTX (24 GB). SMILI and EHT-Imaging are CPU-based methods whose runtime can't be reliably compared. For our method, *DDPM* denotes a DDPM inner loop, while *renoise* variants use the one-step sampler from Section 5.3.

| Methods | PSNR ($\uparrow$) | Time |
|---|---|---|
| *Traditional Methods* | | |
| SMILI (Chandra et al., 2018) | 22.67 (3.13) | – |
| EHT-Imaging (Chael et al., 2018) | 24.28 (3.63) | – |
| *Test-time Guidance Methods* | | |
| DPS (Chung et al., 2023) | 25.86 (3.90) | 24 s |
| LGD (Song et al., 2023b) | 21.22 (3.64) | 36 s |
| RED-diff (Mardani et al., 2024) | 23.77 (4.13) | 16 s |
| PnP-DM (Wu et al., 2024) | 26.07 (3.70) | 45 s |
| DAPS (Zhang et al., 2024) | 25.60 (3.64) | 27 s |
| DiffPIR (Zhu et al., 2023) | 25.01 (4.64) | 21 s |
| *CBG (gradient-free, ours)* | | |
| + DDPM ($K{=}512$) | 26.10 (3.83) | $\sim$ 23 h |
| + renoise ($K{=}128$) | 25.36 (3.99) | 48 s |
| + renoise ($K{=}512$) | 26.10 (3.84) | 2.9 min |
| + renoise ($K{=}4096$) | 26.39 (3.81) | 22.9 min |
| + renoise ($K{=}65536$) | **27.08** (3.74) | $\sim$ 6 h |

of an image $x$ with a given observation $y$. The task is to reconstruct the image from that measurement.

We use the gradient-free estimator in Equation (20), as we find the gradient-based estimator to require many samples. To evaluate the estimator, we compare sampling $x \sim p(x \mid x_t)$ using the pretrained diffusion model with a fixed number of inner steps $M$ and using the renoise trick (compare Section 5.3). For the latter, we train a mean-flow model (Lu et al., 2026) using samples from the original diffusion model. See Appendix H for details.

Table 2 shows that our method matches the state of the art at similar wall-clock time, and significantly outperforms them as the number of per-step samples $K$ is increased. Figure 4 visualizes uncurated qualitative reconstruction results. While some of the baselines produce results that are either noticeably blurred or fail to faithfully follow the ground-truth, our method yields results that are visually consistent with the ground truth. This confirms that our method scales well to high-dimensional scientific inverse tasks.

## 6.3. Image Inverse Problems

In the domain of natural images, the generation of visually pleasing results that score high in likelihood is often the primary goal, while distributional aspects like compliance with the true posterior are secondary. Still, as a proof of concept, we conduct qualitative experiments demonstrating that the proposed method also applies to the image domain. Specifically, we consider two tasks: (a) super-resolution,

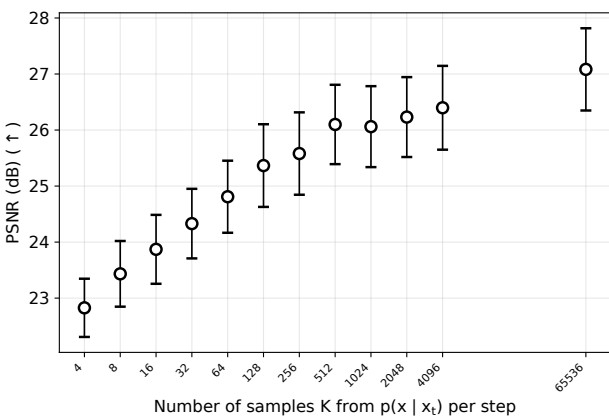

*Figure 5.* Increasing the number of candidate $x$ samples $K$ per denoising step in the gradient-free CBG estimator consistently increases PSNR (higher is better) on black hole imaging. Error bars are 95% confidence intervals over 100 tasks.

and (b) prompt alignment. For super-resolution, we consider a $4\times$ upscaling setting, reconstructing $256 \times 256$ images from $64 \times 64$ low-resolution inputs on natural images from the ImageNet test set (Deng et al., 2009). We use a pixel mean flow model (Lu et al., 2026) trained on ImageNet, and compute the likelihood by measuring the difference between the low-resolution input and the downsampled version. For prompt alignment, we consider a scenario in which a pretrained SANA model (Xie et al., 2025) is prompted to generate images with precise object counts specified in the text prompt. To compute the likelihood, we use a pretrained vision-language reward model (Wang et al., 2025), following the setup of FMTT (Sabour et al., 2025). Detailed experimental settings are described in Appendix I.

As shown in Figure 6, in the super-resolution task, CBG successfully reconstructs the high-resolution ground truth images. In the prompt alignment task, our method consistently generates images with the correct object count, whereas the DPS baseline (Chung et al., 2023) results in suboptimal outcomes. These highlight the practical benefit of CBG in image-domain inverse problems.

## 7. Conclusion

In this work, we address an important gap in the literature of test-time guidance for diffusion models: The common approximations for the diffused likelihoods as well as gradient rescaling can be biased, preventing sampling from the true Bayesian posterior. This may not pose problems for applications such as natural image inverse synthesis, where maximal adherence to the likelihood is preferred over calibrated inference in a Bayesian sense. For scientific scenarios however, proper uncertainty calibration is essential.

We propose two novel estimators that allow arbitrary re-

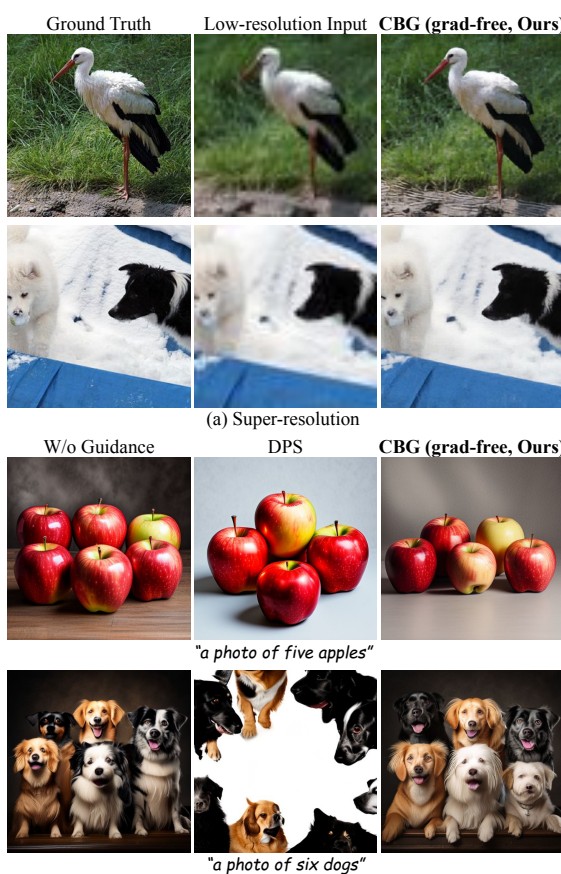

*Figure 6.* We show that CBG is also applicable in the image domain. (a) For image super-resolution, the proposed method effectively reconstructs high-resolution images ($256 \times 256$) from the low-resolution inputs ($64 \times 64$). (b) For prompt alignment, we consider text-to-image generation with exact object-count constraints. Our method satisfies the prompt while preserving image quality, demonstrating the effectiveness of calibrated guidance. In contrast, DPS (Chung et al., 2023) produces incorrect counts or distorted images.

duction of this bias by leveraging the strong signal from evaluating the likelihood function on actual diffusion samples. By increasing computational resources, our method converges to the true Bayesian posterior. The gradient-free estimator is particularly easy to adapt to new scenarios as it does not require computing likelihood and diffusion model gradients.

**Limitations.** Our method requires $p(y \mid x)$ to overlap with the prior in measure; if it concentrates on a measure-zero manifold, samples from $p(x \mid x_t)$ will almost surely miss it. In such cases, methods that over-optimize the likelihood can outperform our calibrated sampler on task metrics. Projecting $x \sim p(x \mid x_t)$ onto the manifold of high likelihood can be a potential solution (Sorrenson et al., 2024), but is out of scope for this work.

## Acknowledgements

Stephan Mandt acknowledges funding from the National Science Foundation (NSF) through an NSF CAREER Award IIS-2047418, IIS-2007719, the NSF LEAP Center, and the Hasso Plattner Research Center at UCI, and the Chan Zuckerberg Initiative. This project was supported by the Chan Zuckerberg Initiative, and the Hasso Plattner Research Center at UCI.

We thank Prakhar Srivastava, Kushagra Pandey and Justus Will for their valuable feedback.

## Impact Statement

This paper presents work whose goal is to advance the field of Machine Learning. There are many potential societal consequences of our work, none which we feel must be specifically highlighted here.

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

# A. Proofs

## A.1. Proof for Theorem 4.1

**Assumption A.1** (Additional condition for Theorem 4.1). Fix $0 < t < 1$, and let $\mu_t(x_t) = \mathbb{E}[x \mid x_t]$. Define

$$M = \sup_{x \in \text{supp } p} p(y \mid x), \qquad S = \{x \in \text{supp } p : p(y \mid x) = M\}. \tag{24}$$

Assume that $S \cap \mu_t(\mathbb{R}^d) \neq \emptyset$ and $\mathbb{P}_{x \sim p(x)}[x \in S] < 1$.

For example, if $\text{supp } p$ is compact, $x \mapsto p(y \mid x)$ is continuous, and the likelihood attains its maximum at an interior point that lies in $\mu_t(\mathbb{R}^d)$, then this assumption is fulfilled.

**Detailed restatement.** Under Assumption A.1, for every $0 < t < 1$, unless the likelihood $p(y \mid x)$ is constant in $x$, there exists some $x_t \in \mathbb{R}^d$ at which the posterior mean approximation in Equation (7) fails.

*Proof.* If the likelihood $p(y \mid x)$ is constant in $x$, then

$$p(y \mid x_t) = \int p(x \mid x_t)p(y \mid x)dx = \int p(x \mid x_t)p(y \mid x = 0)dx = p(y \mid x = 0)\int p(x \mid x_t)dx = p(y \mid x = 0). \tag{25}$$

We now prove the theorem by contradiction under Assumption A.1, assuming that equality holds in Equation (7) for all $x_t \in \mathbb{R}^d$.

First, the diffusion posterior $p(x_t \mid x)$ forms an exponential family:

$$p(x \mid x_t) \propto p(x) \exp\left(-\frac{1}{2b_t^2}\|x_t - a_t x\|^2\right). \tag{26}$$

With the natural parameter $\eta = \frac{a_t}{b_t^2}x_t$ and $h(x) = p(x)\exp(-a_t^2/(2b_t^2)\|x\|^2)$, we get the exponential family:

$$f_X(x \mid \theta) = h(x)\exp(\eta \cdot x - A(\eta)). \tag{27}$$

Since the Gaussian kernel is strictly positive, $p(x \mid x_t)$ and $p(x)$ have the same support for all $x_t \in \mathbb{R}^d$. The natural parameter $\eta$ is a bijection of $x_t$ (on $0 < t < 1$), and so the exponential family is minimal. Therefore, the function $\mu_t(x_t) = \mathbb{E}_{x \sim p(x \mid x_t)}[x] = \nabla_\eta A(\eta)$ is a bijection onto its image (Wainwright & Jordan, 2008, Proposition 3.2).

Let $x^\star \in S \cap \mu_t(\mathbb{R}^d)$. By bijectivity onto the image, there exists $x_t^\star \in \mathbb{R}^d$ with $\mu_t(x_t^\star) = x^\star$. Since $x^\star \in S$, it holds that $p(y \mid x^\star) = M$.

Under our contradiction assumption, Equation (7) holds at $x_t^\star$, and so:

$$p(y \mid x_t^\star) = p(y \mid x = \mu_t(x_t^\star)) = p(y \mid x^\star) = M. \tag{28}$$

On the other hand,

$$p(y \mid x_t^\star) = \int p(x \mid x_t^\star)p(y \mid x)dx. \tag{29}$$

Since $\mathbb{P}_{x \sim p(x)}[x \in S] < 1$, the set

$$A = \text{supp } p \setminus S = \{x \in \text{supp } p : p(y \mid x) < M\} \tag{30}$$

has positive prior probability. Since $p(x \mid x_t^\star) \propto p(x)p(x_t^\star \mid x)$ and the Gaussian factor $p(x_t^\star \mid x)$ is strictly positive for all $x$, the set $A$ also has positive posterior probability under $p(x \mid x_t^\star)$. Therefore,

$$p(y \mid x_t^\star) = \int p(x \mid x_t^\star)p(y \mid x)dx < \int p(x \mid x_t^\star)Mdx = M, \tag{31}$$

contradicting the previous equality. Hence Equation (7) must fail at $x_t^\star$.

$\square$

## A.2. Derivation of Example 4.2

Consider the one-dimensional symmetric two-component Gaussian mixture prior

$$p(x) = \tfrac{1}{2}\mathcal{N}(x; -\mu, 1) + \tfrac{1}{2}\mathcal{N}(x; \mu, 1), \qquad \mu > 0, \tag{32}$$

and the Gaussian likelihood slice centered at the symmetry point

$$p(y = 0 \mid x) = \mathcal{N}(0; x, \rho^2), \qquad \rho^2 > 0. \tag{33}$$

Fix $t \in (0, 1)$, let $s = 1 - t$, and set

$$\sigma_t^2 = \frac{t^2}{s^2 + t^2}, \qquad d = \frac{t^2}{s^2 + t^2}\mu > 0. \tag{34}$$

We claim that the Gaussian proxy identity fails already at $x_t = 0$.

At $x_t = 0$, symmetry gives posterior weights $\tfrac{1}{2}, \tfrac{1}{2}$, and the posterior of each component is Gaussian with variance $\sigma_t^2$ and mean $\pm d$. Hence

$$p(x \mid x_t = 0) = \tfrac{1}{2}\mathcal{N}(x; -d, \sigma_t^2) + \tfrac{1}{2}\mathcal{N}(x; d, \sigma_t^2). \tag{35}$$

Its posterior mean is therefore

$$\mathbb{E}[X \mid X_t = 0] = 0. \tag{36}$$

So the Gaussian proxy is

$$\mathcal{N}(x; 0, \sigma_t^2). \tag{37}$$

Now use the Gaussian convolution identity

$$\int \mathcal{N}(0; x, \rho^2)\mathcal{N}(x; \nu, \sigma_t^2)dx = \mathcal{N}(0; \nu, \rho^2 + \sigma_t^2). \tag{38}$$

Therefore the exact posterior expectation equals

$$\tfrac{1}{2}\mathcal{N}(0; -d, \rho^2 + \sigma_t^2) + \tfrac{1}{2}\mathcal{N}(0; d, \rho^2 + \sigma_t^2) = \mathcal{N}(0; d, \rho^2 + \sigma_t^2), \tag{39}$$

whereas the proxy expectation equals

$$\mathcal{N}(0; 0, \rho^2 + \sigma_t^2). \tag{40}$$

Since $d > 0$, the Gaussian density $\mathcal{N}(0; \nu, \rho^2 + \sigma_t^2)$ is strictly maximized at $\nu = 0$, so

$$\mathcal{N}(0; d, \rho^2 + \sigma_t^2) < \mathcal{N}(0; 0, \rho^2 + \sigma_t^2). \tag{41}$$

Hence

$$\int p(y = 0 \mid x)p(x \mid x_t = 0)dx \neq \int p(y = 0 \mid x)\mathcal{N}\left(x; \mathbb{E}[X \mid X_t = 0], \sigma_t^2\right)dx. \tag{42}$$

Thus the Gaussian posterior proxy fails for this example.

## A.3. Proof for Theorem 4.3

**Detailed restatement.** Fix $0 < t < 1$ and $\gamma \in \mathbb{R}_+ \setminus \{0, 1\}$. If there is no constant $c \in \mathbb{R}_+$ such that $p(y \mid x) = c$ for $p(x)$-almost every $x$, then there exists some $x_t \in \mathbb{R}^d$ at which the naive tempered diffused likelihood Equation (12) fails.

*Proof.* Let $\phi(u) = u^\gamma$ for $u \geq 0$. For $\gamma \in (0, 1)$, $\phi$ is strictly concave; for $\gamma > 1$, $\phi$ is strictly convex.

For fixed $x_t$, define $x \sim p(x \mid x_t)$ and $z = p(y \mid x) \in [0, \infty)$. Then

$$\int p(x \mid x_t)p(y \mid x)^\gamma dx = \left(\int p(x \mid x_t)p(y \mid x)dx\right)^\gamma \tag{43}$$

is exactly $\mathbb{E}[\phi(z)] = \phi(\mathbb{E}[z])$. By Lemma A.2, this equality holds if and only if $z$ is almost surely constant under $p(x \mid x_t)$.

Assume now that the naive identity Equation (12) holds for all $x_t$. Fix any $x_t^\star \in \mathbb{R}^d$. Then there exists $c \in \mathbb{R}_+$ such that $p(y \mid x) = c$ for $p(x \mid x_t^\star)$-almost every $x$. Since

$$p(x \mid x_t^\star) \propto p(x) \exp\left(-\frac{\|x_t^\star - a_t x\|^2}{2b_t^2}\right), \tag{44}$$

and the Gaussian factor is strictly positive, $p(x \mid x_t^\star)$ and $p(x)$ have the same null sets. Hence

$$p(y \mid x) = c \qquad p(x)\text{-almost surely}. \tag{45}$$

Therefore, if no such constant $c$ exists $p(x)$-almost surely, the naive identity cannot hold for all $x_t$. Hence there must exist some $x_t$ at which Equation (12) fails. $\qquad\square$

**Lemma A.2.** *Let $\phi : [0, \infty) \to \mathbb{R}$ be strictly convex or strictly concave, and let $Z$ be an integrable random variable with values in $[0, \infty)$ such that $\phi(Z)$ is integrable. Then*

$$\mathbb{E}[\phi(Z)] = \phi(\mathbb{E}[Z]) \quad \Longleftrightarrow \quad Z \text{ is almost surely constant}. \tag{46}$$

*Proof.* If $Z$ is almost surely constant, the identity is immediate.

We now prove the converse. First assume that $\phi$ is strictly convex and that $Z$ is not almost surely constant. Let $m = \mathbb{E}[Z]$ and define the events

$$A = \{Z < m\}, \qquad C = \{Z \geq m\}. \tag{47}$$

Since $Z$ is not almost surely constant and $m$ is its mean, both $\mathbb{P}(A)$ and $\mathbb{P}(Z > m)$ are positive. In particular, with $\lambda = \mathbb{P}(A) \in (0, 1)$,

$$m_- = \mathbb{E}[Z \mid A], \qquad m_+ = \mathbb{E}[Z \mid C] \tag{48}$$

satisfy $m_- < m < m_+$ and

$$m = \lambda m_- + (1 - \lambda)m_+. \tag{49}$$

Applying Jensen's inequality conditionally on $A$ and $C$, we obtain

$$\mathbb{E}[\phi(Z)] = \lambda\,\mathbb{E}[\phi(Z) \mid A] + (1 - \lambda)\,\mathbb{E}[\phi(Z) \mid C] \tag{50}$$

$$\geq \lambda\,\phi(m_-) + (1 - \lambda)\,\phi(m_+) \tag{51}$$

$$> \phi(\lambda m_- + (1 - \lambda)m_+) \tag{52}$$

$$= \phi(m). \tag{53}$$

The strict inequality uses strict convexity and $m_- \neq m_+$. This contradicts $\mathbb{E}[\phi(Z)] = \phi(\mathbb{E}[Z])$.

Therefore, if $\phi$ is strictly convex, equality can only hold when $Z$ is almost surely constant. If $\phi$ is strictly concave, then $-\phi$ is strictly convex, and applying the previous argument to $-\phi$ gives the same conclusion. $\qquad\square$

# B. Algorithm

---
**Algorithm 1** Calibrated Bayesian Guidance
---
**Input:** Initial noise $x_T \sim \mathcal{N}(0, 1)$
**for** $t = T$ **to** $1$ **do**
    Draw samples from $p(x \mid x_t)$
    Compute $\nabla_{x_t} \log p(x_t \mid y)$ with Equations (5) and (16) or Equation (20)
    Compute $x_{t-1}$ with an Euler step along Equation (2)
**end for**
---

## C. Gradient-Free and Gradient-Based Methods

We consider the case where we have a simple prior $p(x) = \mathcal{N}(x; 2, 1)$ and a simple likelihood $p(y|x) = \mathcal{N}(y; 0, 0.4^2)$. Denote $\hat{x}(x_t)$ as the exact analytic estimate for $\mathbb{E}[x|x_t, y]$, and $\hat{x}_{\text{grad-free}}(x_t), \hat{x}_{\text{grad-based}}(x_t)$ our estimates from the CBG gradient-free and CBG gradient-based algorithms using Tweedie's formula. We plot the variances $\mathbb{E}\left[(\hat{x}_{\text{grad-based}}(x_t) - \hat{x}(x_t))^2\right]$ and $\mathbb{E}\left[(\hat{x}_{\text{grad-free}}(x_t) - \hat{x}(x_t))^2\right]$ as we vary $x_t$ and $t$. Both estimators use $K = 1000$ samples, with the variance expectation estimated over 100 algorithm calls. Looking at Figure 7, there is a clear region to the left of $t \approx 0.3$ where the gradient-based method performs better, but noticeably not by much.

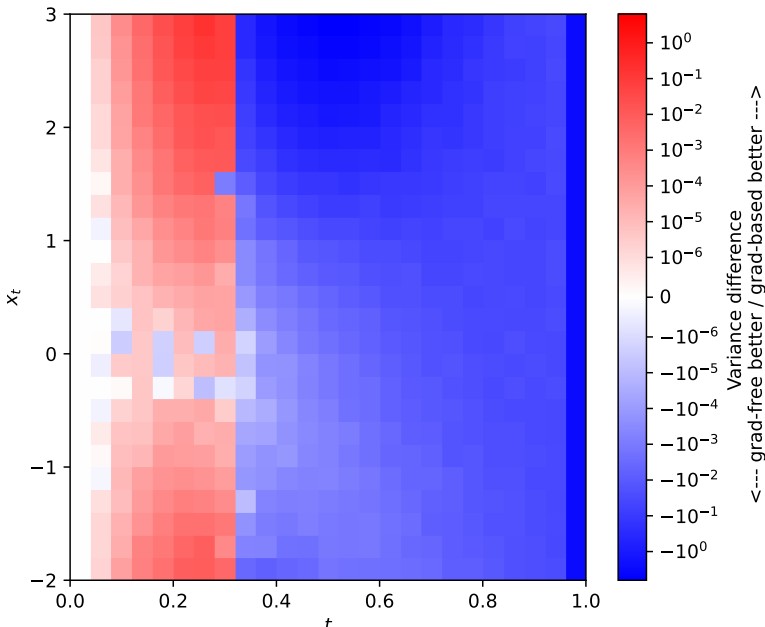

*Figure 7.* Variance comparison for gradient-free and gradient-based methods. In some regions, the gradient-based method has lower variance.

## D. Bias and Variance of Diffusion Methods

We consider Task 4 of the Bayesian Inference Benchmark, specifically estimating $\mathbb{E}[x \mid x_t, y]$. We initialize an $x_t$ from $p(x_t|y, t = 0.5)$ and run one step of each diffusion sampling algorithm to estimate $\mathbb{E}[x \mid x_t, y]$, with varying values for $K$. We repeat this process 100 times to estimate the bias and variance of each method compared to an analytic solution. The results are shown in Figure 8. We see that all methods converge to 0 variance as $K$ increases, but only our CBG methods (both gradient-free and gradient-based) converge to the correct analytic solution. Notably, the gradient-free version of CBG converges much faster than the gradient-based version, which is consistent with the variance comparison in Figure 7.

## E. Bayesian Inference Benchmark Guidance Methods

### E.1. Diffusion Process

For all guidance methods, the flow matching forward process was used:

$$p(x_t|x) = \mathcal{N}(x_t; (1 - t)x, t^2 I)$$

For generation, $t$ is scheduled to linearly decrease from 1 to 0.

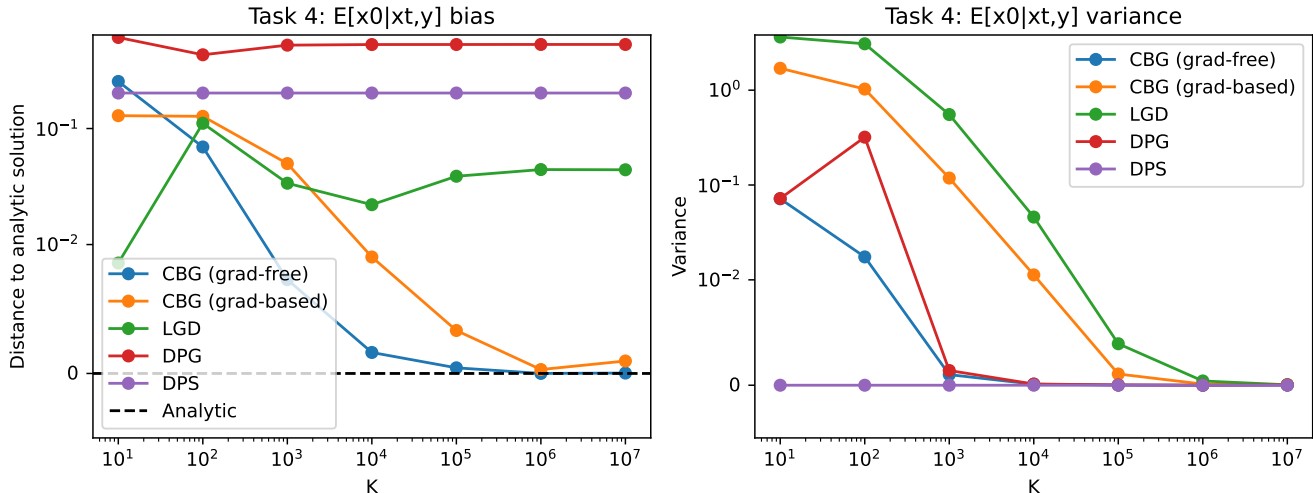

*Figure 8.* Bias and variance of $\mathbb{E}[x \mid x_t, y]$ as a function of $K$ for Bayesian Inference Task 4. While the variance of all methods decreases to 0 with more compute, only our CBG methods decrease to 0 bias.

### E.2. Diffusion Posterior Sampling

Diffusion Posterior Sampling (DPS) (Chung et al., 2023) approximates the score $\nabla_{x_t} \log p(y|x_t)$ as

$$\nabla_{x_t} \log p(y|x_t) \approx \nabla_{x_t} \log p\left(y|\mathbb{E}\left[x|x_t\right]\right)$$

### E.3. Loss Guided Diffusion

Loss Guided Diffusion (LGD) (Song et al., 2023b) approximates the score $\nabla_{x_t} \log p(y|x_t)$ as

$$\nabla_{x_t} \log p(y|x_t) \approx \nabla_{x_t} \log \int p(y|x)q(x|x_t)dx$$

where $q(x|x_t) = \mathcal{N}\left(\mathbb{E}\left[x|x_t\right], \frac{t^2}{(1-t)^2+t^2}I\right)$ approximates the true diffusion posterior distribution. Evaluated using a finite set of samples $x^{(i)} \sim q(x|x_t)$ for $i = 1, \cdots, K$, we have:

$$\nabla_{x_t} \log p(y|x_t) \approx \nabla_{x_t} \log \left[\frac{1}{K}\sum_i p\left(y|x^{(i)}\right)\right]$$

### E.4. Diffusion Policy Gradient

Diffusion Policy Gradient (DPG) (Tang et al., 2023) approximates the score $\nabla_{x_t} \log p(y|x_t)$ as

$$\nabla_{x_t} \log p(y|x_t) \approx C\frac{s(x_t)}{\|s(x_t)\|_2^2}$$

Thus:

$$s(x_t) = \mathbb{E}_{x \sim q(x|x_t)}\left[\left(p(y|x) - b\right)\nabla_{x_t}\left(-\|x - \mathbb{E}\left[x|x_t\right]\|_2^2\right)\right]$$

where $q(x|x_t) = \mathcal{N}\left(\mathbb{E}\left[x|x_t\right], \frac{t^2}{(1-t)^2+t^2}I\right)$ approximates the true diffusion posterior distribution. Evaluated using a finite set of samples $x^{(i)} \sim q(x|x_t)$ for $i = 1, \cdots, K$, we approximate as:

$$s(x_t) \approx \frac{1}{K}\sum_i \left(p(y|x^{(i)}) - b^{(i)}\right)\nabla_{x_t}\left(-\|x^{(i)} - \mathbb{E}\left[x|x_t\right]\|_2^2\right)$$

where $b^{(i)}$ is computed as

$$b^{(i)} = \frac{1}{K-1} \sum_{j=1, j \neq i}^{K} p\left(y | x^{(j)}\right)$$

## E.5. Stochastic Control Guidance

Stochastic Control Guidance (SCG) (Huang et al., 2024) chooses an $\hat{x}_{t'}$ to transition to, given the current state $x_t$, as follows:

$$\hat{x}_{t'} = \arg\max_{x_{t'}} p\left(y | \mathbb{E}[x | x_{t'}]\right)$$

Approximating $p(x_{t'} | x_t)$ as

$$q(x_{t'} | x_t) = \mathcal{N}\left(\frac{t'}{t} x_t + \left(1 - \frac{t'}{t}\right) \mathbb{E}[x | x_t], \sigma^2 I\right), \sigma^2 = \frac{(t - t')^2}{(1 - t)^2 + t^2}$$

we can evaluate this using a finite set of samples $x_{t'}^{(i)} \sim q(x | x_t)$ for $i = 1, \cdots, K$ as:

$$\hat{x}_{t'} \approx \arg\max_{i} p\left(y | \mathbb{E}\left[x | x_{t'}^{(i)}\right]\right)$$

## E.6. Langevin + Noisy Classifier

Langevin + Noisy Classifier performs annealed Langevin Sampling (Song & Ermon, 2019) on $p(x_t)p(y|x = x_t)$, where a noisy image $x_t$ is passed to the classifier $p(y|x)$.

## E.7. Langevin + REINFORCE

Langevin + REINFORCE performs annealed Langevin Sampling (Song & Ermon, 2019) using an estimate of $\nabla_{x_t} \log p(x_t \mid y)$ generated by the REINFORCE estimator in our CBG gradient-free algorithm (20).

## E.8. MCMC No-U-Turn Sampler

MCMC No-U-Turn Sampler uses the No-U-Turn Sampler sampler from (Hoffman & Gelman, 2014) to sample directly on the analytic unnormalized distribution $p(x|y) \propto p(x)p(y|x)$.

# F. Bayesian Inference Benchmark Guidance Tasks

## F.1. Task 1

Inference on the mean of a 10-dimensional Gaussian, with a Gaussian prior.

**Prior**: $p(x) = \mathcal{N}(x; 0, 0.1I)$

**Diffusion Posterior**: $p(x \mid x_t) = \mathcal{N}\left(x; \frac{1-t}{10t^2 + (1-t)^2} x_t, \frac{t^2}{10t^2 + (1-t)^2} I\right)$.

**Likelihood**: $p(y|x) = \mathcal{N}(x; y, 0.1I)$

**Dimensionality**: $x \in \mathbb{R}^{10}$

## F.2. Task 2

Inference on the mean of a 10-dimensional Gaussian, with a uniform prior.

**Prior**: $p(x) = \mathcal{U}(-1, 1)$

**Diffusion Posterior**: $p(x \mid x_t) \propto \mathbf{1}\{x \in [-1, 1]^{10}\} \mathcal{N}\left(x; \frac{x_t}{1-t}, \frac{t^2}{(1-t)^2} I\right)$.

**Likelihood**: $p(y|x) = \mathcal{N}(y; x, 0.1I)$

**Dimensionality**: $x \in \mathbb{R}^{10}, y \in \mathbb{R}^{10}$

### F.3. Task 3

Inference on the parameters of a 10-dimensional Gaussian with nonlinear mean and variance functions, with a uniform prior.

**Prior**: $p(x) = \mathcal{U}(-3, 3)$

**Diffusion Posterior**: $p(x \mid x_t) \propto \mathbf{1}\{x \in [-3,3]^5\} \mathcal{N}\left(x; \; \frac{x_t}{1-t}, \; \frac{t^2}{(1-t)^2} I\right).$

**Likelihood**: $p(y|x) = \prod_{i=1}^{4} \mathcal{N}\left(y^{(i)}; \mu(x), \Sigma(x)\right)$

where $\mu(x) = \begin{bmatrix} x_1 \\ x_2 \end{bmatrix}, \Sigma(x) = \begin{bmatrix} s_1^2 & \rho s_1 s_2 \\ \rho s_1 s_2 & s_2^2 \end{bmatrix}, s_1 = x_3^2, s_2 = x_4^2, \rho = \tanh x_5$

**Dimensionality**: $x \in \mathbb{R}^5, y \in \mathbb{R}^{4 \times 2}$

### F.4. Task 4

Inference on the shared mean of a mixture of two 2-dimensional Gaussians, one much broader than the other, with a uniform prior.

**Prior**: $p(x) = \mathcal{U}(-10, 10)$

**Diffusion Posterior**: $p(x \mid x_t) \propto \mathbf{1}\{x \in [-10, 10]^2\} \mathcal{N}\left(x; \; \frac{x_t}{1-t}, \; \frac{t^2}{(1-t)^2} I\right).$

**Likelihood**: $p(y|x) = 0.5\mathcal{N}\left(y; x, I\right) + 0.5\mathcal{N}\left(y; x, 0.01I\right)$

**Dimensionality**: $x \in \mathbb{R}^2, y \in \mathbb{R}^2$

### F.5. Task 5

Inference on a two-moons distribution with both bimodality and local structure, with a uniform prior.

**Prior**: $p(x) = \mathcal{U}(-1, 1)$

**Diffusion Posterior**: $p(x \mid x_t) \propto \mathbf{1}\{x \in [-1,1]^2\} \mathcal{N}\left(x; \; \frac{x_t}{1-t}, \; \frac{t^2}{(1-t)^2} I\right).$

**Likelihood**: Let $s(x) = \begin{bmatrix} -\frac{|x_1+x_2|}{\sqrt{2}} \\ \frac{-x_1+x_2}{\sqrt{2}} \end{bmatrix}, z = y - s(x), u = z_1 - 0.25, v = z_2, \rho = \sqrt{u^2 + v^2}, \phi = \mathrm{atan2}(v, u)$. Then, for $\rho > 0$,

$$p(y \mid x) = \mathbf{1}\left\{\phi \in \left(-\frac{\pi}{2}, \frac{\pi}{2}\right)\right\} \cdot \frac{1}{\pi \rho} \cdot \frac{\mathcal{N}(\rho; 0.1, 0.01^2)}{\Phi(10)}.$$

## G. Experimental Details for Bayesian Inference Benchmark

Table 3 contains the set of hyperparameters tried for all methods, and table 4 contains the chosen hyperparameters to maximize distributional fit. All combinations of hyperparameters were tried, restricting to combinations satisfying $N \cdot K \leq 10^5$, which restricts the total number of likelihood evaluations. CBG does not tune a guidance scale since we aim to sample from the true distribution $p(x|y)$ at $\gamma = 1$. Additionally, some other diffusion models do not use the maximum number of likelihood evaluations since they converge to the wrong distribution, and happen to get better results with less evaluations.

$10^4$ reference samples are given, and the methods generate $10^4$ samples as well.

For experimental details on likelihood-free methods, see Lueckmann et al. (2021).

## H. Experimental Details for Black Hole Imaging

We evaluate two configurations of CBG on the black-hole benchmark of InverseBench (Zheng et al., 2025). In both cases, we use the gradient-free estimator from Equation (20). We tune hyperparameters on the validation split and report all final numbers on 100 test samples. For baselines, we adopt the hyperparameter settings reported in Table 12 of the original

*Table 3.* Bayesian inference benchmark hyperparameter sweep.

| Methods | $N$ | $K$ | $\gamma$ | $C$ |
|---|---|---|---|---|
| CBG (gradient-free) | $[10^1, 10^5]$ | $[10^1, 10^5]$ | N/A | N/A |
| CBG (gradient-based) | $[10^1, 10^5]$ | $[10^1, 10^5]$ | N/A | N/A |
| DPS | $[10^1, 10^5]$ | N/A | $[10^{-3}, 10^3]$ | N/A |
| LGD | $[10^1, 10^5]$ | $[10^1, 10^5]$ | $[10^{-3}, 10^3]$ | N/A |
| DPG | $[10^1, 10^5]$ | $[10^1, 10^5]$ | $[10^{-3}, 10^3]$ | $[10^{-1}, 10^1]$ |
| SCG | $[10^1, 10^5]$ | $[10^1, 10^5]$ | $[10^{-3}, 10^3]$ | N/A |

*Table 4.* Bayesian inference benchmark chosen hyperparameters.

| Methods | Task 1 | | | | Task 2 | | | | Task 3 | | | | Task 4 | | | | Task 5 | | | |
|---|---|---|---|---|---|---|---|---|---|---|---|---|---|---|---|---|---|---|---|---|
| | $N$ | $K$ | $\gamma$ | $C$ | $N$ | $K$ | $\gamma$ | $C$ | $N$ | $K$ | $\gamma$ | $C$ | $N$ | $K$ | $\gamma$ | $C$ | $N$ | $K$ | $\gamma$ | $C$ |
| CBG (grad-free) | 100 | 1000 | – | – | 100 | 1000 | – | – | 100 | 1000 | – | – | 100 | 1000 | – | – | 100 | 1000 | – | – |
| CBG (grad-based) | 100 | 1000 | – | – | 100 | 1000 | – | – | 1000 | 100 | – | – | 1000 | 100 | – | – | 100 | 1000 | – | – |
| DPS | 100 | – | 0.022 | – | 100 | – | 0.022 | – | 100 | – | 0.005 | – | 100 | – | 0.005 | – | 10000 | – | 0.001 | – |
| LGD | 100 | 1000 | 2.154 | – | 10 | 100 | 10.000 | – | 10 | 1000 | 0.464 | – | 10 | 10 | 0.100 | – | 10 | 10000 | 0.464 | – |
| DPG | 10 | 1000 | 10.000 | 10.000 | 10 | 1000 | 2.154 | 10.000 | 100 | 100 | 0.001 | 0.100 | 100 | 100 | 0.464 | 1.000 | 100 | 10 | 215.443 | 10.000 |
| SCG | 10 | 10 | – | – | 100 | 10 | – | – | 10 | 100 | – | – | 10 | 100 | – | – | 10 | 100 | – | – |

InverseBench paper to ensure fair comparison.

For the *full* setting in Table 2, we use the pretrained diffusion model released with InverseBench to sample from $p(x \mid x_t)$ with a DDPM inner loop. The outer denoising process uses $N = 1000$ steps. We select the inner-loop length as $M = 50$ and the number of per-step samples based on validation performance. We use guidance scale $\gamma = 0.003$ for the reported full-model result. Table 2 reports the TITAN RTX runtime used for the main comparison.

As Table 2 shows, the compute time with the pretrained DDPM model is still too long to be practical. We therefore additionally evaluate the renoising strategy from Section 5.3 and Equation (21) using a one-step mean-flow model. Because the training dataset for the black-hole images is not publicly released, we use the pretrained diffusion model above to generate $\sim 140\,000$ synthetic training samples, and train a mean-flow model (Lu et al., 2026) on this dataset. It has the same architecture as the DDPM model.

In this *renoise* setting, proposal generation requires one model evaluation per sample instead of a 50-step DDPM inner loop, plus one model call to denoise $\hat{x} = f_t(x_t)$. We also reduce the outer denoising process to $N = 100$, which we found sufficient in practice, and sweep different values of $K$. The rows labeled *renoise* in Table 2 report this $K$-sweep, and the additional PSNR and $\chi^2$ ablations in Figures 5 and 9 use the same setup.

## I. Experimental Details for Image Inverse Problems

**Super-resolution.** We target $4\times$ super-resolution on ImageNet (Deng et al., 2009) as a stress test of the Calibrated Bayesian Guidance (CBG) estimator on a high-dimensional inverse problem. For each ground-truth $256\times256$ image we form a $64\times64$ observation by bicubic downsampling; the likelihood is the Gaussian $y \sim \mathcal{N}\big(\text{bicubic}_{4\times}(x),\, \sigma_y^2 I\big)$ with $\sigma_y = 0.01$, matching the SR3 benchmark used by Pandey et al. (2025b) and the baselines below. Super-resolution then reduces to drawing from the Bayesian posterior with this likelihood and an ImageNet prior.

While guidance methods for ImageNet inverse problems typically use the classifier-guided diffusion prior of Dhariwal & Nichol (2021), the number of inner diffusion steps required for each candidate makes the overall budget prohibitive at the candidate counts CBG requires. We instead take the pixel mean flow (pMF) model of Lu et al. (2026) as the prior; specifically, the released class-conditional pMF-B/16 checkpoint at $256\times256$. The model is one-step, and we can generate candidates with the fast renoising procedure of Section 5.3.

Because pMF-B/16 is class-conditional but super-resolution is not, we feed the bicubic LR through the noise-aware $64\times64$ ImageNet classifier of Dhariwal & Nichol (2021) (evaluated at $t{=}0$) and condition pMF on the classifier's top-1 prediction. Classifier-free guidance is used with $\omega = 3.0$.

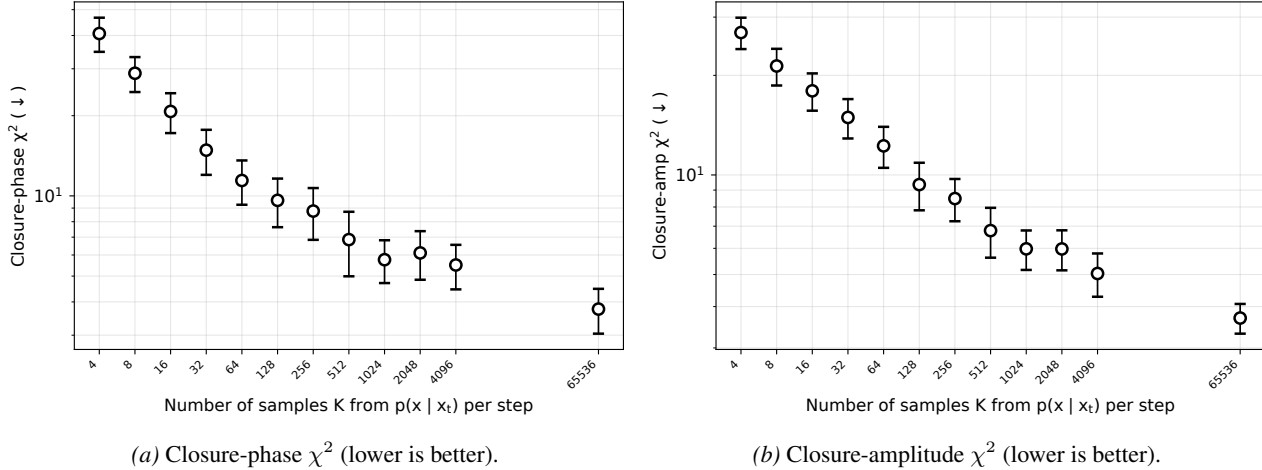

*(a)* Closure-phase $\chi^2$ (lower is better).

*(b)* Closure-amplitude $\chi^2$ (lower is better).

*Figure 9.* $\chi^2$ data-consistency metrics for the $K$-ablation in Figure 5. Both scale log-log in $K$ and improve monotonically past $K = 512$. Markers show per-$K$ means with 95% confidence intervals over 100 tasks; red lines are log-log OLS fits to the means.

*Table 5.* $4\times$ super-resolution on ImageNet ($256\times256$), evaluated on the 1000-image SR3 test list of Pandey et al. (2025b) (likelihood noise $\sigma_y$=0.01). Paper rows are reproduced from Pandey et al. (2025b) (Tables 2 and 12); FID/KID are computed with clean-fid in `legacy_pytorch` mode, matching the paper's evaluation code. Best per column in bold.

| Method | PSNR↑ | SSIM↑ | LPIPS↓ | FID↓ | KID↓ |
|---|---|---|---|---|---|
| **Ours (pMF tiled gradient-free CBG with renoise)** | **24.41** | 0.668 | 0.298 | 40.74 | 0.0033 |
| DPS | 23.61 | 0.676 | 0.195 | 30.67 | 0.0021 |
| DDRM | 24.15 | **0.701** | 0.325 | 52.76 | 0.0151 |
| RED-diff | 24.06 | 0.685 | 0.354 | 51.83 | 0.0084 |
| C-ΠGDM | 23.20 | 0.631 | 0.270 | 39.96 | 0.0024 |
| MPGD | 21.83 | 0.587 | 0.215 | 37.39 | 0.0017 |
| RB-Modulation | 23.41 | 0.674 | 0.211 | 35.26 | 0.0032 |
| NDTM | 23.12 | 0.674 | **0.158** | **28.75** | **0.0011** |

While CBG can be applied directly to super-resolution with the per-image MSE log-likelihood as the reward, that reward signal is very sparse: a single scalar per candidate must summarise the agreement between a full $256\times256$ image and the LR observation. The gradient-free estimator then struggles to reconstruct fine detail unless compute is increased dramatically, because most candidates contribute essentially the same global reward and the per-pixel direction is averaged out.

Intuitively, think of an image such as the White Stork in Figure 10 with the stork in the center, grass behind, and mud below. It is unlikely that a single sampled candidate will sample all three regions in a single composition; far more often, different candidates recover different parts. The job of the estimator is to recombine these partially correct candidates into one globally correct direction.

Fortunately, the SR MSE loss decomposes as a sum of local pixel losses, so we can compute and weight the estimator on a per-tile basis. For $4\times$ SR from $64\times64$ to $256\times256$ we use one tile per LR pixel, so each tile covers a $4\times4$=16-pixel patch of the HR image, giving $64\times64$=4096 tiles in total. For each tile we evaluate the per-tile log-likelihood of every candidate and compute a tile-local softmax-weighted target; the global drift direction is then the soft stitch of these per-tile targets. This decoupling lets entirely different weightings be used for different regions of the image and is what makes the estimator usable on SR at a competitive compute budget.

Table 5 reports our run on the 1000-image SR3 list of Pandey et al. (2025b). CBG was run with $K = 2048$ candidates per step, 8 outer denoising steps, and $64\times64$ tiles, for a total of $16\,384$ pMF evaluations per image. Super-resolving a single image took on average $74\,\text{s}$ on 4 workstation-class GPUs (NVIDIA Titan RTX). Our run leads in PSNR and is competitive on SSIM, but trails the strongest perceptual baselines (NDTM, DPS, RB-Modulation) on LPIPS and FID. The fact that the prior model needs to be different between our method and the baselines, and that the number of diffusion steps is very

different, remains as a limitation to compare the numbers.

**Prompt alignment.** We use the pretrained SANA-1600M model (Xie et al., 2025) with the classifier-free guidance scale of 4.5 and generate images with resolution of $512 \times 512$. We use a VLM-based reward function computed with the pretrained Skywork-VL-Reward-7B model (Wang et al., 2025). For each text prompt, we construct a corresponding binary visual question that checks whether the generated image satisfies the prompt-specific constraint. For instance, for the prompt "a photo of six dogs," we ask: "Hint: Please answer the question with exactly one word, Yes or No. Question: Does this image depict exactly six dogs? Criteria: - Count all visible dogs in the image - The total number must be exactly six (no more, no less) Choices: (A) no (B) yes". Following FMTT (Sabour et al., 2025), we define the reward function value as $\sigma(\text{logits}[\text{Yes}] - \text{logits}[\text{No}])$, where $\sigma(\cdot)$ denotes the sigmoid function. We set $N = 20$, $M = 20$, $K = 256$, and select $\gamma$ in the range $[100, 500]$.

## J. Additional Analysis for Black Hole Imaging

Figure 9 shows the additional analysis of the relationship between $K$ and performance, in terms of $\chi^2$-distance metric. We also visualize several samples for two random black hole imaging tasks in Figures 11 and 12. Figure 13 shows five samples that were generated by the mean-flow variant with renoise.

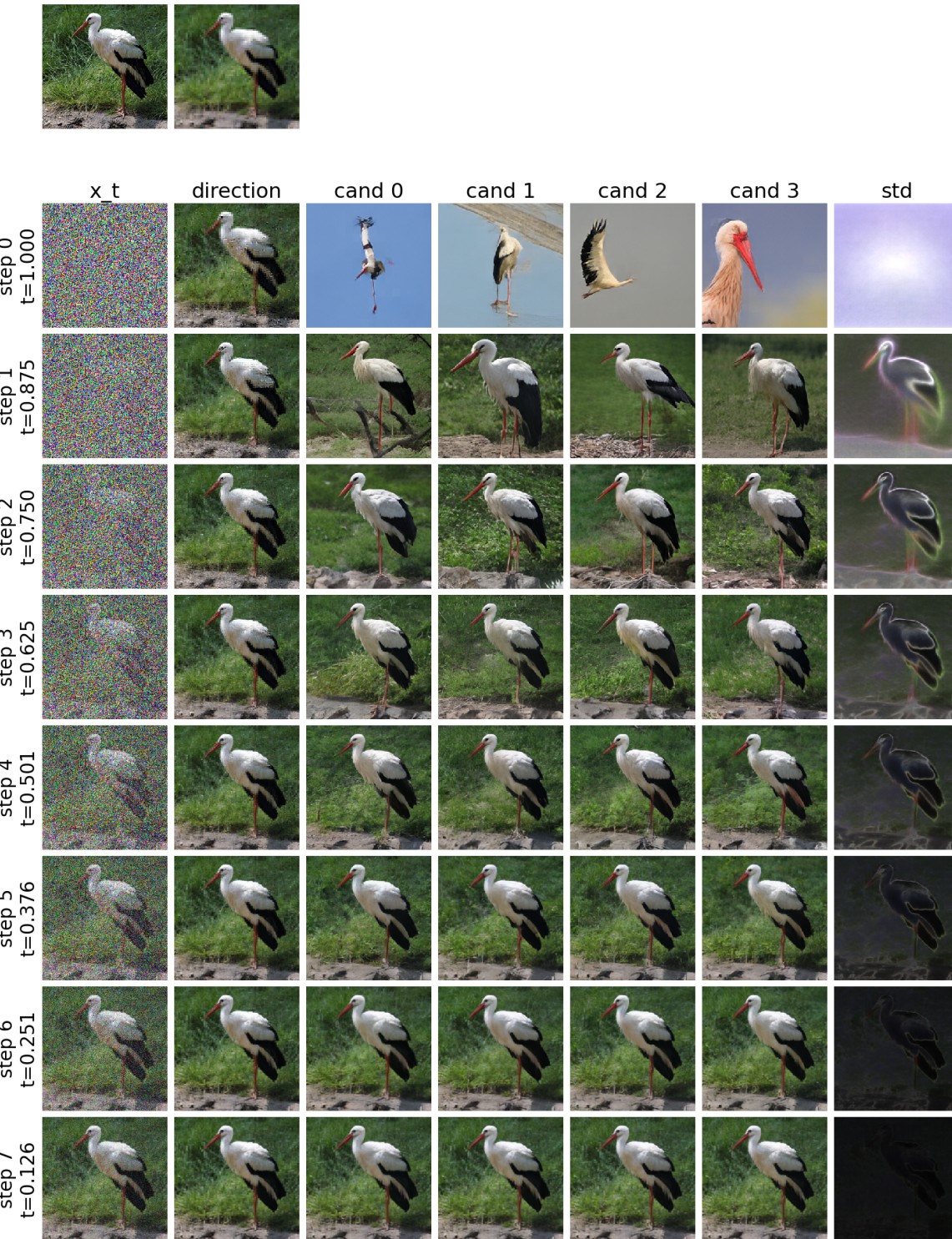

*Figure 10.* Super-resolution of an image of a White Stork. The ground truth image is $256 \times 256$ and gets downscaled to $64 \times 64$. For each of the eight denoising steps, four out of the 2048 candidates are shown. Out of these candidates, the tiled gradient-free estimator assembles a "direction", and the noise is updated towards it. It is visible that the tiled gradient-free estimator can assemble an image early that follows the down-scaled version. The fact that the direction provides a strong signal allows for only 8 denoising steps. The rightmost column visualizes the per-pixel standard deviation of the proposals, where a brighter color channel signals higher diversity. Consequently, the area will be blue if the blue color channel disagrees, white if all disagree, and black when the candidates agree on that pixel. As the denoising process progresses, the samples are highly diverse, but converge towards the final image.

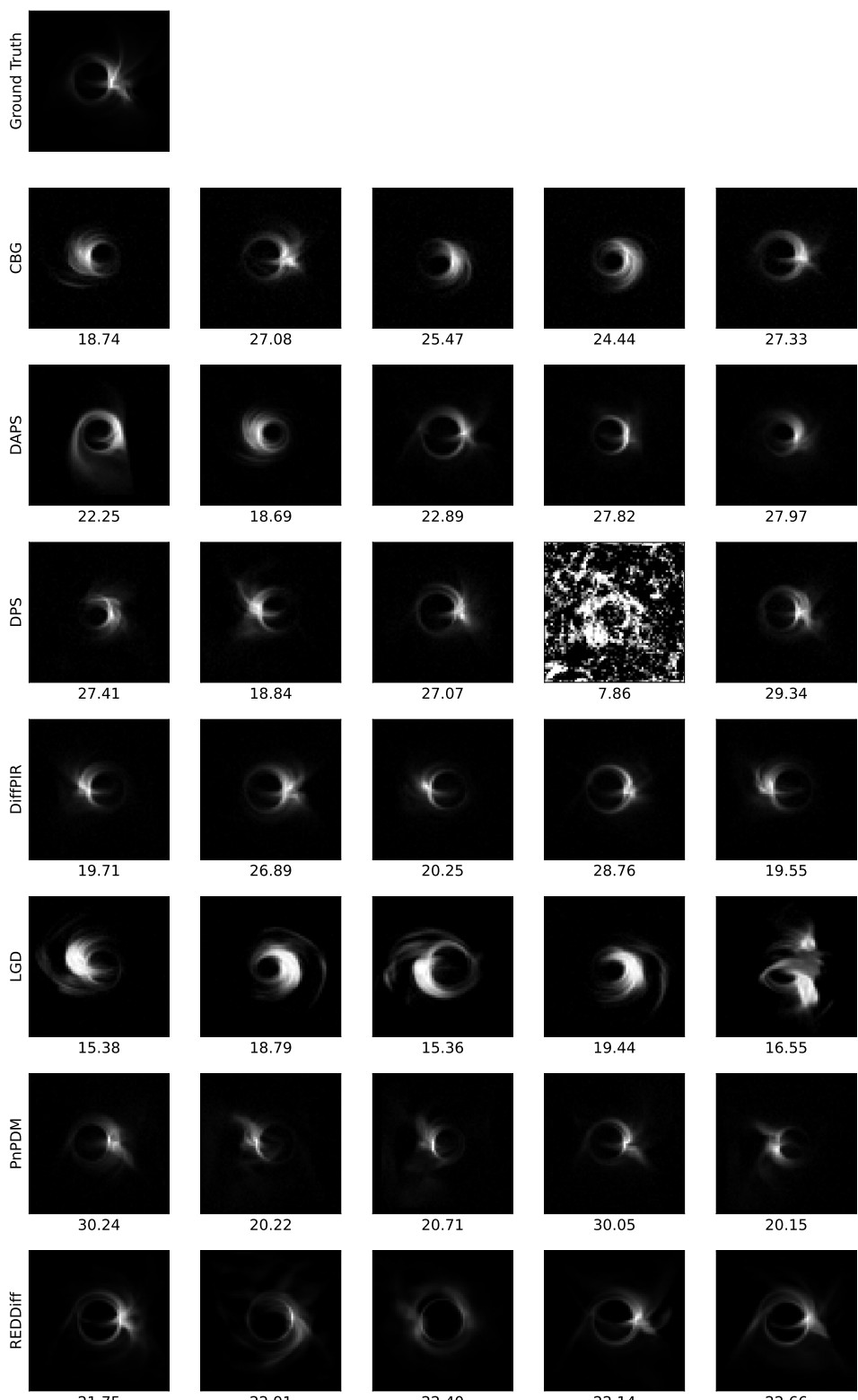

*Figure 11.* Qualitative analysis of the generative results of the different methods with a random index, five samples were taken per method. The numbers are the PSNR to the ground truth for each task. CBG is the implementation based on the pretrained diffusion model (1000 steps outer loop, 50 steps inner loop, 512 particles)

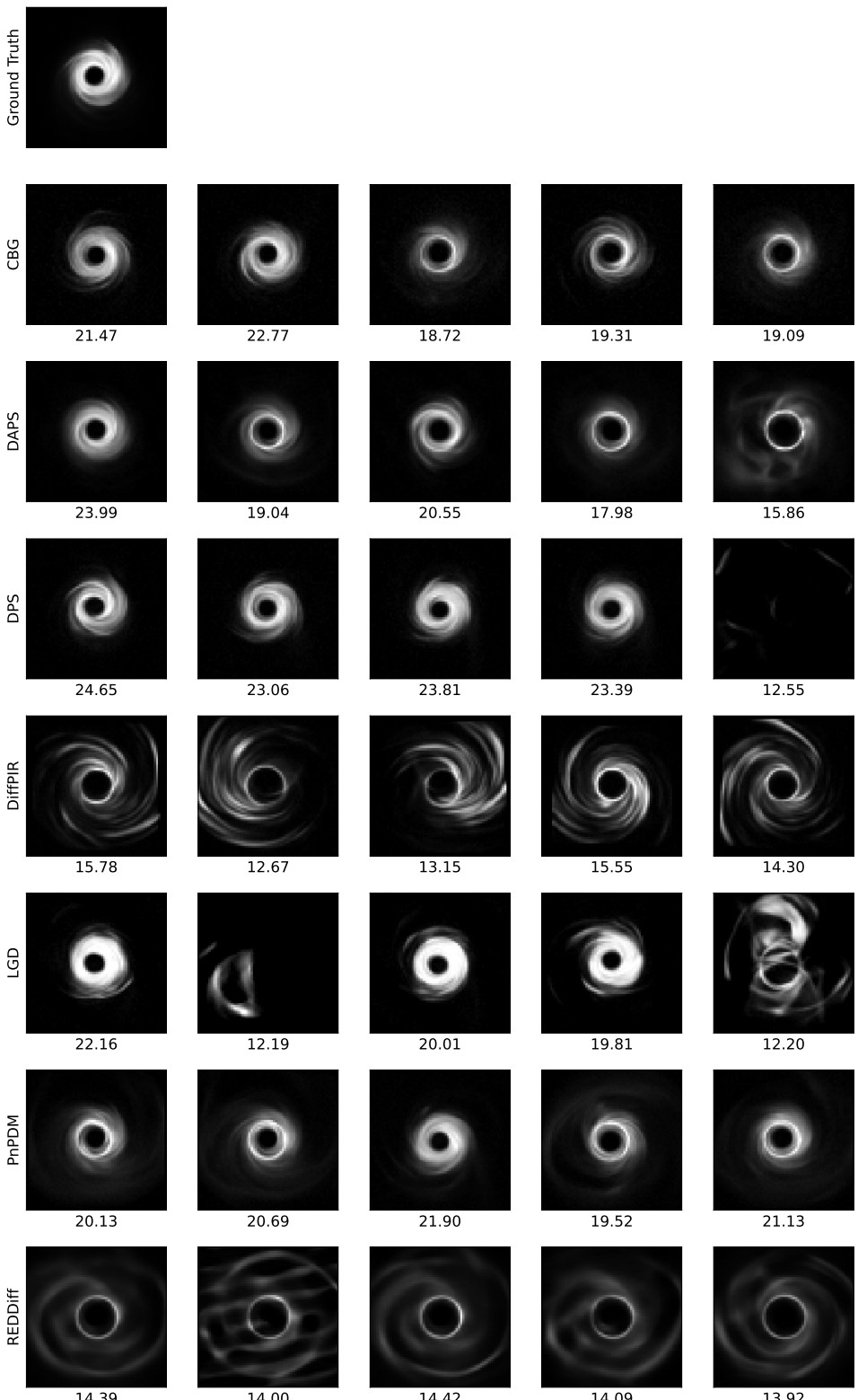

*Figure 12.* Qualitative analysis of the generative results of the different methods with a different random index, five samples were taken per method. The numbers are the PSNR to the ground truth for each task. CBG is the implementation based on the pretrained diffusion model (1000 steps outer loop, 50 steps inner loop, 512 particles)

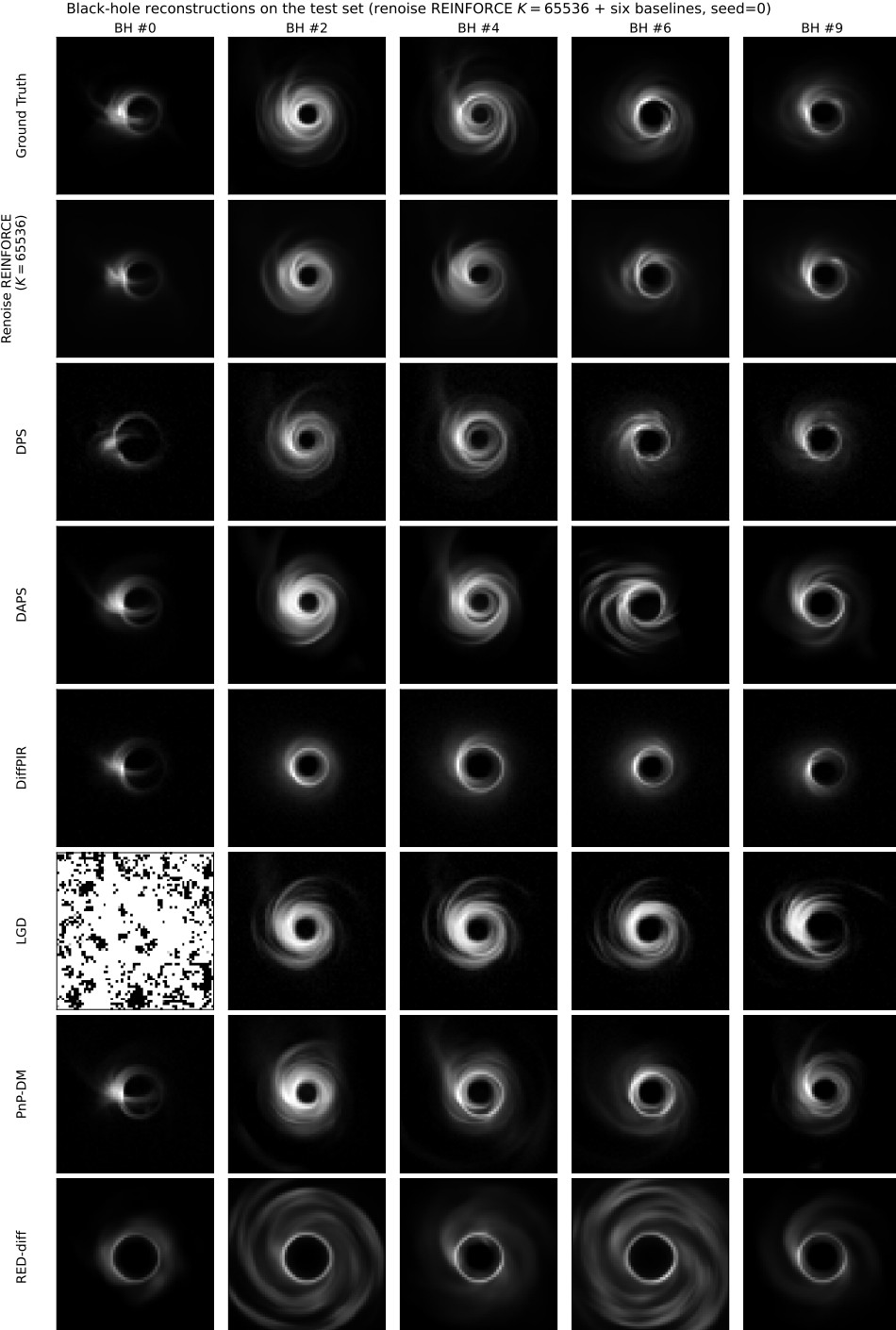

*Figure 13.* Blackhole reconstruction with the gradient-free pixel mean flow model compared to the baselines. Reconstructions were done with $K = 65\,536$

