# OpenReview forum: "Calibrated Test-Time Guidance for Bayesian Inference"
_ICML.cc/2026/Conference — ICML 2026 regular_

### Official Review · Reviewer_Qqm4 · 2026-03-08

**Soundness:** 3
**Presentation:** 4
**Significance:** 2
**Originality:** 2
**Overall Recommendation:** 5
**Confidence:** 4

**Summary:**

The paper first analyses that existing guidance mechanisms lead to biased estimates of the posterior distributions. To mitigate this they propose a new framework for both differentiable and non differentiable rewards.They establish the competitive performance of their method on toy examples as well as a black hole imaging case.

**Compliance With Llm Reviewing Policy:**

Affirmed.

**Final Justification:**

I was a bit concerned with the technical novelty, but after the rebuttal, the new improvements on the theoretical results and reading a bit more on the literature I've decided to increase my score

**Key Questions For Authors:**

1. The method requires sampling from p(x | x_t) which can substantially increase. Apart from the performance gains it will be useful to have a comparison in terms of computational budget with previous biased approaches.

2. How does the method scale with dimension? Did you observe weight degeneracy as dimension grows?

3. In Theorem 4.1 the lipschitzness assumption on p(x) is fairly strong. Is there any way that this can be relaxed?

4. Is it possible to leverage the ideas proposed with biased method to attain a middle ground in terms of computational cost and performance improvement?

**Limitations:**

yes

**Strengths And Weaknesses:**

Strengths:
- The paper is very well written and easy to follow.
- The methodology developed relies on a detailed theoretical analysis.
- The problem addressed in the paper of getting samples from a tilted distribution having access to samples from the prior is very relevant and the paper provides a new direction for obtaining unbiased samples that could be used for downstream tasks.

Weaknesses:
- The new introduced method does not provide a big technical novelty. The proposed framework for obtaining unbiased samples comes with a big increase in computational cost without significant performance gains.

---

> ### Author Rebuttal · Authors · 2026-03-31
>
> We thank the reviewer for the helpful feedback!
>
> > The new introduced method does not provide a big technical novelty. The proposed framework for obtaining unbiased samples comes with a big increase in computational cost without significant performance gains.
>
> We think that Theorems 4.1, 4.2 and 4.3 are highly relevant for the community, as they directly pinpoint relevant issues that need to be addressed when applying guidance methods to scientific inference, where unbiased sampling is critical.
>
> We agree that our proposed method comes at a computational burden, but our evidence suggests that this is necessary to get unbiased samples. As we increase the number of samples for different methods, we find that only our method produces an unbiased estimate for the gradient of the noisy likelihood. We illustrate this for an intermediate step of Task 4 in [this figure](https://imgur.com/a/0CRo1Kd).
>
> > The method requires sampling from $p(x | x_t)$ which can substantially increase [the required computational budget]. Apart from the performance gains it will be useful to have a comparison in terms of computational budget with previous biased approaches.
>
> There are two cost factors: the inner diffusion loop, and the number of samples $K$ required for a reliable estimate. Theorems 4.1 and 4.2 say that we need samples from $p(x \mid x_t)$ to get unbiased samples, and the estimators in Eq. (16) and (22) converge to the true value with more samples, so neither cost can be fully avoided.
>
> For the Bayesian inverse tasks in Table 1, $p(x \mid x_t)$ is available analytically, so the inner loop cost does not apply and wall-clock times across all methods are identical. For the black hole task (Table 2), our method requires around 2h40 on a single GPU, about 70x times longer than PnP-DM, the second best method. As we mention in the Limitations section, this can be reduced by two orders of magnitude by learning a diffusion posterior one-shot model $p(x \mid x_t)$, reducing the time to lower than PnP-DM. We admit that this is a weakness of the proposed method, but it does sample from the correct posterior, something that none of the baselines can guarantee (see Theorems 4.1 and 4.2).
>
> As for the cost of scaling $K$, we find performance to increase logarithmically: PSNR ≈ 1.2 ln(K) + 18.4, so most of the gain over biased baselines is recovered at modest $K$, and the cost can be continuously traded off against quality.
>
> > How does the method scale with dimension? Did you observe weight degeneracy as dimension grows?
>
> General statements are difficult here, as performance depends on the task: specifically, how concentrated the posterior is relative to the prior as a function of dimension. Per the request of reviewers 8gNk and G5Md, we add a qualitative high-dimensional image inverse problem (8192-dimensional latent space for 512×512 images), using CBG with a vision-language reward model to better adhere to an image composition prompt. Since image composition does not change as dimension increases, the task itself does not become harder with dimensionality: the model only needs to add finer details, which are unspecified by the likelihood. However, if the reward function's scaling causes the posterior to shrink relative to the prior (for example if the likelihood looks at high-frequency details such as image sharpness), the task would become harder. This applies equally to all methods.
>
> With regards to the weight degeneracy, for the 4096-dimensional blackhole task, we find that only few samples adhere to the likelihood early in sampling as the diversity of samples is too high, [see this figure](https://imgur.com/a/8qHmjIS). This results in an average weight of the most likely sample of around 97% for most of the sampling. Later, alternative solutions achieve higher likelihood, with the most likely sampling dropping to 41% in the last steps.
>
> > In Theorem 4.1 the lipschitzness assumption on $p(x)$ is fairly strong. Is there any way that this can be relaxed?
>
> Yes, thanks for the question! Based on your suggestion, we revisited Theorems 4.1 and 4.2 and made two improvements: (1) We drop the Lipschitz constraint, and (2) we generalize the statement to hold for every $1>t>0$ (instead of just claiming that there is some $t$ for which there is an error).
>
> Thanks again for the suggestion!
>
>
> > Is it possible to leverage the ideas proposed with biased method to attain a middle ground in terms of computational cost and performance improvement?
>
> Yes, one can replace the inner diffusion loop for fewer steps to obtain a sample. For example, DPS is recovered by running zero diffusion steps and taking the Tweedie estimate. An interpolation could run a few inner diffusion steps to get diversity, and then take the posterior mean prediction. In the blackhole experiments, we run the inner diffusion with larger step sizes to reduce computation.
>
> We hope this addresses the reviewer’s concerns, and we look forward to the upcoming discussion.

---

> > ### Author Rebuttal · Reviewer_Qqm4 · 2026-04-03
> >
> > Thanks for the detailed response. I really appreciate the relaxation of the assumptions needed for your main theorems. Based on your response I've updated my score

---

> > > ### Author Response · Authors · 2026-04-03
> > >
> > > We thank the reviewer for the positive response and continued interest in our work!

---

### Official Review · Reviewer_XrX9 · 2026-03-09

**Soundness:** 3
**Presentation:** 3
**Significance:** 3
**Originality:** 3
**Overall Recommendation:** 4
**Confidence:** 3

**Summary:**

The framework of test-time guidance for diffusion models is developed with the principles of the Bayes rule. However, the usage of test-time guidance for Bayesian inference is not well-explored. This work considers the task of generating posterior samples from a statistical model where the prior distribution is a diffusion model. Existing works in test-time guidance do not lead to the correct Bayesian posterior, even after increasing the computational budget, because of the inherent bias from approximating an integral. To address the issue, this work derives two unbiased estimates of the guidance gradients that lead to the correct posterior. One estimate comes from the reparameterization trick, which is called Gradient-Based Calibrated Bayesian Guidance. The other works similar to the REINFORCE estimator, which is called Gradient-Free Calibrated Bayesian Guidance. It is found that the latter is more suitable for the task of getting diffusion posterior and scales better with dimension. The experiments verify that the existing approaches generate biased posteriors, while the proposed ones have reduced error with more computation. On a benchmark for simulation-based inference and a model for black-hole imaging, it is demonstrated that the proposed approaches generate better posteriors than existing baselines.

**Compliance With Llm Reviewing Policy:**

Affirmed.

**Final Justification:**

I would like to thank the authors for updating with additional experiments, especially the MCMC comparisons. I have increased my score accordingly.

**Key Questions For Authors:**

- How does changing $K$ affect the posterior sampling?
- What is the running time of different methods in the benchmarking and the black-hole imaging experiments?
- How does the proposed methods compare with MCMC-based posterior sampling?

**Limitations:**

yes

**Strengths And Weaknesses:**

Strengths:
- The problem of getting a correct posterior from a pretrained diffusion model is important. The proposed methods are sound, and can be applied in many fields with a diffusion-based foundation model.
- The ideas in this work are novel, and well connected to the literature.
- The work is also well-written and has a clear guidance of how it can be used.

Weaknesses:
- There is a hyperparameter $K$ to control the gradient estimation variance. Intuitively a larger $K$ will lead to a better posterior estimation. However, I do not see any experiments of varying $K$.
- For the experiments in Table 1 and Table 2, the running time of different methods are not provided. The likelihood-free methods should also produce the correct posterior, given enough data. It will be interesting to see if the proposed methods have a better utilization of the computational budget.
- I also do not understand why there are no likelihood-based methods in Table 1, since the likelihood is assumed to have a closed-form expression. In this case, MCMC should be a suitable method for sampling from the posterior distribution.
- (minor) Calibration has a special meaning in probabilistic forecasting. I would call the method "corrected test-time guidance" instead. But the current name should not be an issue if its relationship with the existing calibration literature is properly stated.

---

> ### Author Rebuttal · Authors · 2026-03-31
>
> We thank the reviewer for the helpful feedback!
>
> > There is a hyperparameter $K$ to control the gradient estimation variance. Intuitively a larger K will lead to a better posterior estimation. [...] How does changing $K$ affect the posterior sampling?
>
> The reviewer’s intuition is correct: We ran an additional experiment evaluating the bias and variance of our CBG estimators. We find that only for our estimators, the variance of the estimator going to zero indicates that the bias vanishes, too. The other methods do not converge to the correct estimate, resulting in biased samples.
>
> See [this plot for the Bayesian Inference tasks in Table 1](https://imgur.com/a/0CRo1Kd) and [this plot for the blackhole images](https://imgur.com/a/ld2FIjb) for confirmation.
>
>
> > For the experiments in Table 1 and Table 2, the running time of different methods are not provided. The likelihood-free methods should also produce the correct posterior, given enough data. It will be interesting to see if the proposed methods have a better utilization of the computational budget. [...] What is the running time of different methods in the benchmarking and the black-hole imaging experiments?
>
> For Table 1, runtimes are identical between our method and the other diffusion methods, and on the order of milliseconds. This is because the diffusion posterior is known in closed form, avoiding the inner diffusion loop. For the likelihood-free methods, there is an additional amortization cost of training a separate generative model for the likelihood.
>
> For Table 2, our method requires around 2h40 for a single image on a single GPU.
>
> > I also do not understand why there are no likelihood-based methods in Table 1, since the likelihood is assumed to have a closed-form expression. In this case, MCMC should be a suitable method for sampling from the posterior distribution.
> > How does the proposed methods compare with MCMC-based posterior sampling?
>
> We outperform MCMC on all but one task. Here is MCMC added to Table 1, and our method for reference:
>
> | Method | Task 1 | Task 2 | Task 3 | Task 4 | Task 5 | Avg |
> |---|---|---|---|---|---|---|
> | MCMC | 0.514 (0.005) | 0.511 (0.004) | 0.627 (0.005) | 0.512 (0.007) | 0.603 (0.046) | 0.553 |
> | Grad-free CBG (ours) | **0.505 (0.004)** | 0.513 (0.009) | 0.584 (0.063) | **0.507 (0.006)** | **0.525 (0.028)** | **0.527** |
>
> MCMC has the disadvantage that it has a hard time crossing low-likelihood regions, evidenced by the relatively high C2ST in tasks 3 and 5. Note that we can only run MCMC reliably for the Bayesian Inference task because the prior is given by a closed-form analytic expression. This is not directly compatible with diffusion priors learned from data, *e.g.* the blackhole imaging task in our paper.
>
>
> > (minor) Calibration has a special meaning in probabilistic forecasting. I would call the method "corrected test-time guidance" instead. But the current name should not be an issue if its relationship with the existing calibration literature is properly stated.
>
> Thanks for the comment, we will clarify the relationship with the calibration literature.
>
> We hope this addresses the reviewer’s concerns, and we look forward to the upcoming discussion.

---

> > ### Author Rebuttal · Reviewer_XrX9 · 2026-03-31
> >
> > I would like to thank the authors for the detailed rebuttal. I will increase my score given that the new results are incorporated into the revision. Regarding the new MCMC baseline, what is the MCMC method used here?

---

> > > ### Author Response · Authors · 2026-04-01
> > >
> > > We thank the reviewer for the positive response and continued interest in our work!
> > >
> > > > Regarding the new MCMC baseline, what is the MCMC method used here?
> > >
> > > Stan’s default implementation of MCMC was used, specifically the No-U-Turn Sampler (NUTS). This is an adaptive variant of Hamiltonian Monte Carlo that also makes use of gradients. We used 4 chains and a long adaptive warmup period of 10^5 samples.

---

### Official Review · Reviewer_G5Md · 2026-03-12

**Soundness:** 2
**Presentation:** 3
**Significance:** 2
**Originality:** 3
**Overall Recommendation:** 3
**Confidence:** 3

**Summary:**

The paper shows that usual diffusion guidance methods do not sample from the actual Bayesian posterior and proposes a new method called the Calibrated Bayesian Guidance (CBG).

**Compliance With Llm Reviewing Policy:**

Affirmed.

**Final Justification:**

The authors have carefully explained the computational bottleneck of the method in the paper. This addresses my concerns to a good extent and in response I am updating my score.

**Key Questions For Authors:**

What is the variance vs compute trade-off of the CBG versus approximate baselines as dimension grows?

**Limitations:**

Yes

**Strengths And Weaknesses:**

**Strengths**
* Distributional testing shows CBG improves toward optimal with more likelihood evaluations, unlike usual approximation based guidance.


**Weaknesses**
* CBG estimators would increase compute substantially due to Monte Carlo sampling compared to usual approximation based guidance methods.
* The paper lacks large scale diffusion model experiments on realistic data showing the computational over head is manageable for significant gain in quality.

---

> ### Author Rebuttal · Authors · 2026-03-31
>
> We thank the reviewer for the helpful feedback!
>
> > CBG estimators would increase computation substantially due to Monte Carlo sampling compared to usual approximation based guidance methods.
>
> There are two cost factors: the inner diffusion loop, and the number of samples $K$ required for a reliable estimate. Theorems 4.1 and 4.2 say that we need samples from $p(x \mid x_t)$ to get unbiased samples, and the estimators in Eq. (16) and (22) converge to the true value with more samples, so neither cost can be fully avoided.
>
> However, the cost of the inner diffusion loop can be avoided by training a conditional one-step model that learns $p(x \mid x_t)$ (see our Limitations section). This is effectively what we do for the Bayesian inverse tasks in Table 1 since $p(x \mid x_t)$ is available analytically, and results in the wall-clock times between the different guidance methods are identical.
>
> As for the cost of scaling $K$, we find performance to increase logarithmically as $K$ increases for the black hole data: we find PSNR ≈ 1.2 ln(K) + 18.4, see [this plot](https://imgur.com/a/ld2FIjb). This means practitioners can tune $K$ to their compute budget and still obtain substantially better-calibrated samples than any biased baseline.
>
>
>
> > The paper lacks large scale diffusion model experiments on realistic data showing the computational overhead is manageable for significant gain in quality.
>
> Thank you for the valuable comment. We emphasize that the core contribution of our work is theoretical: we identify that existing methods sample from an incorrect distribution and propose CBG as a principled correction that samples from the correct distribution. Existing methods underperform due to the miscalibrated sampling, whereas our method significantly outperforms them, demonstrating that it samples much closer to the true distribution.
>
> We are happy to confirm in a qualitative experiment that CBG also works in the image domain and show the results in [this figure](https://imgur.com/a/3URuS9g). We consider images requiring precise semantic alignment with text prompt (*e.g.* “a photo of *five* apples”), where diffusion models are known to fail at correctly reflecting attributes such as object count. For loss calculation, we use the pretrained vision-language reward model, following the setup of FMTT (https://arxiv.org/abs/2511.22688). Using the pretrained SANA model (https://huggingface.co/Efficient-Large-Model/Sana_1600M_512px_diffusers), our method consistently generates images with the correct count, while baselines do not, highlighting the practical benefit of calibrated guidance.
>
>
>
> > What is the variance vs compute trade-off of the CBG versus approximate baselines as dimension grows?
>
> Thanks for the interesting question! We ran an additional experiment evaluating the bias and variance of our CBG estimators. We find that only for our estimators, the variance of the estimator going to zero indicates that the bias vanishes, too. The other methods do not converge to the correct estimate, resulting in biased samples.
>
> In detail: All sample-based estimators need more computation as the dimension increases, since it becomes harder to get high likelihood samples. However, this is required to get an unbiased estimate for a sample from the true posterior. Our paper, in particular Theorems 4.1 and 4.2 indicate that this may be a fundamental barrier for calibrated Bayesian posterior inference. Please [see this plot](https://imgur.com/a/0CRo1Kd) showing the performance of our estimators on Task 4, if we fix a specific $x_t$ and increase $K$, the number of samples used to estimate $E[x_0|x_t, y]$. Additionally, please see the second attached plot showing the performance of our estimators on a variant of Task 4, where we fix a specific $K$ (which is proportional to the compute budget across methods) and increase the dimension.
>
> We hope this addresses the reviewer’s concerns, and we look forward to the upcoming discussion.

---

> > ### Author Rebuttal · Reviewer_G5Md · 2026-04-03
> >
> > I very much appreciate the detailed feedback from the authors, especially related to my concerns about the computation. Hence, I will update my score to weak reject, 3.

---

> > > ### Author Response · Authors · 2026-04-06
> > >
> > > We thank the reviewer for increasing their score. We would highly appreciate if the reviewer could communicate which points are not addressed by our rebuttal, so that we can reach common ground.
> > >
> > > To our understanding, the reviewer fully agrees that (quoted from the review):
> > >
> > > - The paper shows that usual diffusion guidance methods do not sample from the actual Bayesian posterior
> > > - [Our method] CBG improves toward optimal [guidance]
> > >
> > > The concerns of the reviewer were:
> > >
> > > - the computational cost
> > >     - We show that this is the only way known to us to remove the bias in existing methods, and the cost is low enough that we can still solve interesting tasks (see our experiments). Also, at identical compute, we outperform all competitors on the Bayesian inference task. This includes MCMC and Langevin sampling, thanks to suggestions by reviewers 8gNk and Qqm4.
> > > - the lack of high-dimensional experiments.
> > >     - In the rebuttal, we provided a qualitative experiment on prompt adherence in high-dimensional image generation, [see this experiment](https://imgur.com/a/3URuS9g) --- thanks to the reviewer for the suggestion.
> > >
> > > We hope this clarifies the merits of our paper.

---

### Official Review · Reviewer_8gNk · 2026-03-13

**Soundness:** 3
**Presentation:** 3
**Significance:** 1
**Originality:** 2
**Overall Recommendation:** 3
**Confidence:** 1

**Summary:**

The paper identifies an inherent estimation bias in existing Classifier-Free Guidance (CFG) frameworks and proposes a test-time scaling mechanism to improve the accuracy of the guidance signal. By refining the guidance estimation during inference, the method achieves superior generation quality without requiring model retraining.

**Compliance With Llm Reviewing Policy:**

Affirmed.

**Final Justification:**

Please refer to ack.

**Key Questions For Authors:**

* Please cite related literatures.

* Why not directly apply the Langevin Dynamics to narrow the bias like in https://arxiv.org/pdf/1907.05600 , which is more elegant.

**Strengths And Weaknesses:**

**Strengths**
- Clearly identifies and formalizes the bias introduced by CFG.

**Weaknesses**
- The bias itself is not entirely new. Prior work (CEP https://arxiv.org/abs/2304.12824; iCFG https://openreview.net/pdf?id=0QAzIMq32X) has already pointed out related issues, but the paper does not adequately acknowledge or cite that literature.
- The experiments are too toy-like to support the paper’s claims. In the diffusion literature, large-scale image models have been standard for years; even an evaluation on an older model such as Stable Diffusion 1.5 would be much more informative. Results on small toy settings have very limited significance here.
- The computational complexity of the proposed algorithm is not practical. Its cost appears prohibitive, which substantially limits its usefulness.

---

> ### Author Rebuttal · Authors · 2026-03-31
>
> We thank the reviewer for the helpful feedback!
>
> > [Strengths] Clearly identifies and formalizes the bias introduced by CFG.
>
> Thanks for the assessment. We would like to clarify that only Theorem 4.3 concerns classifier-free guidance, and Theorems 4.1 and 4.2 formalize bias in test-time guidance frameworks in the case of likelihood / reward functions.
>
>
> > The bias itself is not entirely new. Prior work (CEP; iCFG) has already pointed out related issues [...]
>
> Thanks for these pointers, we are happy to add these to the paper. We think that our precise characterizations in Theorems 4.1, 4.2 and 4.3 are relevant and novel results. In detail:
>
> Regarding CEP, their paper concerns learning diffused likelihoods effectively, while we consider training-free yet guided inference. In terms of theoretical contributions, they prove that their method learns the diffused likelihood at the optimum. Similarly, our method samples from the unbiased posterior given enough estimator samples, overcoming the problems of other methods proven in Theorems 4.1 and 4.2.
>
> Regarding iCFG & our Theorem 4.3: While Theorem 3.1 in iCFG also concerns CFG, it does not offer the actionable insight in our work: their theorem shows that the enhanced transition kernel generally differs from the original. Our Theorem 4.3 provides a precise necessary and sufficient condition when this has practical consequences: The estimate is biased unless the likelihood function is constant. We propose adding that context to our Theorem 4.3.
>
> Also, due to a suggestion by reviewer Qqm4, we have strengthened our theoretical contributions to remove unnecessary assumptions (Lipschitz) and show that all three cases are biased for every t.
>
> > The experiments are too toy-like to support the paper’s claims. [...]
>
> We respectfully disagree. Our paper’s main claim is that existing classifier free guidance (CFG) methods for diffusion models do not sample from the correct distribution, and we propose a new method that does produce the correct distribution. Table 1 on page 8 shows the distributional fit of our method compared to other existing CFG methods and existing likelihood-free methods not based on diffusion. All other diffusion methods perform extremely poorly on these tasks, indicating that they are sampling from the wrong distribution. Our gradient-free CBG method not only far outperforms existing diffusion methods, but outperforms most likelihood-free methods as well, indicating that we are sampling much closer to the correct distribution. The fact that existing methods are not well-calibrated in a simple experiment, while our methods are, confirms our claims.
>
> We are happy to confirm in a qualitative experiment that CBG also works in the image domain and show the results in [this figure](https://imgur.com/a/3URuS9g). We consider images requiring precise semantic alignment with text prompt (*e.g.* “a photo of *five* apples”), where diffusion models are known to fail at correctly reflecting attributes such as object count. For loss calculation, we use the pretrained VLM, following the setup of FMTT (https://arxiv.org/abs/2511.22688). Using the pretrained SANA model (https://huggingface.co/Efficient-Large-Model/Sana_1600M_512px_diffusers), our method consistently generates images with the correct count, while baselines do not, highlighting the practical benefit of calibrated guidance.
>
> > The computational complexity of the proposed algorithm is not practical. [..]
>
> There are two cost factors: the inner diffusion loop, and the number of samples $K$. Theorems 4.1 and 4.2 say that we need samples from $p(x \mid x_t)$ to get unbiased samples, and the estimators in Eq. (16) and (22) converge to the true value with more samples, so neither cost can be fully avoided.
>
> However, it is worth noting that the cost of the inner diffusion loop can be avoided by training a conditional one-step model that learns $p(x \mid x_t)$ (see our Limitations section). This is effectively what we do for the Bayesian inverse tasks in Table 1 since $p(x \mid x_t)$ is available analytically, and results in the wall-clock times between the different guidance methods are identical.
>
> As for the cost of scaling $K$, we find performance to increase logarithmically as $K$ increases, [see this plot](https://imgur.com/a/ld2FIjb): for the black hole data, we find $PSNR ≈ 1.2 ln(K) + 18.4$. This means practitioners can tune $K$ to their compute budget and still obtain substantially better-calibrated samples than any biased baseline.
>
> > Why not directly apply the Langevin Dynamics [...]
>
> This is an interesting idea, but we think it does not solve the problem: the referenced paper argues that computing the score of the data distribution is difficult, which would lead to biased samples in Langevin sampling. Our method, on the other hand, reliably offers unbiased samples due to consistent estimators.
>
> We hope this addresses the reviewer’s concerns, and we look forward to the upcoming discussion.

---

> > ### Author Rebuttal · Reviewer_8gNk · 2026-04-04
> >
> > 1. I understand your algorithm uses additional inference to reach the true posterior. However, my point is that the CFG-induced bias is not a new discovery; it was identified shortly after CFG was introduced. The authors should properly acknowledge and cite prior research on this bias and cite related work in the first two theorem (maybe).
> >
> > 2. Practically speaking, eliminating this bias introduces substantial overhead. Despite previous claims of bias elimination in the literature, modern large-scale diffusion models largely ignore this and stick to standard CFG simply because it yields the best cost-performance balance.
> >
> > 3. The core issue with your approach is the sheer cost of estimating $p(x \mid x_t)$. For $N$ diffusion steps, this estimation demands roughly $N^2/2$ inference steps, rendering the algorithm highly inefficient. Moreover, learning $p(x \mid x_t)$ is an extremely hard condition to satisfy. In the marginal case, it requires the model to perform one-step generation directly from a Gaussian distribution. This is a significantly harder problem—essentially what Consistency Models are designed to do.
> >
> > 4. I pointed out Langevin Dynamics precisely because it can also converge to the true posterior, but it avoids these expensive and impractical estimations entirely.
> >
> > In summary, my core argument is that the proposed algorithm fundamentally relies on estimations that are either computationally prohibitive or intrinsically harder to learn. Furthermore, this computational and estimation burden is an inherent bottleneck whenever we attempt to scale diffusion models to high-dimensional datasets.
> >
> > But my point is all from the diffusion point, therefore I will change my confidence to 1.

---

> > > ### Author Response · Authors · 2026-04-06
> > >
> > > Thanks for the detailed feedback. Let us clarify our contributions:
> > >
> > > - **Theorems 4.1 and 4.2** describe a fundamental bias in the existing test-time/classifier guidance literature. **Existing methods converge to the wrong solution** with more compute. These methods are therefore inappropriate for high-stakes settings such as scientific inference and medical imaging, where accuracy is more important than computational cost.
> > > - **Calibrated Bayesian Guidance (CBG) converges to the Bayesian posterior** with enough compute. See our Figure 1, as well as the vanishing bias for Bayesian inference ([see this figure](https://imgur.com/a/0CRo1Kd)) and higher quality black hole imaging ([see this figure](https://imgur.com/a/ld2FIjb)).
> > > - This is why **CBG achieves state of the art quality**, comparable or better than strong baselines on Bayesian inference and blackhole imaging (our Tables 1 and 2, as well as new results below).
> > >
> > > We hope this gives the reviewer confidence in confirming the paper's importance.
> > >
> > > ---
> > >
> > > Detailed answer to the reviewer's questions:
> > >
> > > > However, my point is that the CFG-induced bias is not a new discovery; it was identified shortly after CFG was introduced.
> > > > The authors should properly acknowledge and cite prior research on this bias and cite related work in the first two theorems
> > >
> > > For classifier-free guidance (CFG), we agree that the existence of a bias is known (see iCFG paper). The novelty of our Theorem 4.3 is the actionable formulation, aligned with Theorem 4.1 and 4.2. We will make this distinction clear in the camera ready version.
> > >
> > > For classifier guidance, **Theorems 4.1 and 4.2 are novel**; we are not aware of such precise results in prior work. For example, $\Pi$GDM is explicitly motivated to improve approximations to $p(x \\mid x_t)$. However, the bias is not analyzed mathematically (see page 4 of Song et al. ICLR 2023 at https://openreview.net/pdf?id=9_gsMA8MRKQ) -- a gap that our theorems 4.1 and 4.2 fill.
> > >
> > > > [...] eliminating this bias introduces substantial overhead. Despite previous claims of bias elimination in the literature, modern large-scale diffusion models largely ignore this and stick to standard CFG simply because it yields the best cost-performance balance
> > >
> > > Our work focuses on classifier guidance (CG), not classifier-free guidance (CFG), see title and abstract. These are different setups: CFG tries to get the best performance out of a conditionally-trained model, whereas CG assumes that the pretrained diffusion model is fixed and the conditioning is formulated by a likelihood/reward. This is common in scientific inference, where likelihoods are explicit but no data pairs of high and degraded quality are available to train a CFG model.
> > >
> > > > The core issue with your approach is the sheer cost of estimating $p(x \mid x_t)$. For $N$ diffusion steps, this estimation demands roughly $N^2/2$ inference steps, rendering the algorithm highly inefficient. Moreover, learning $p(x \mid x_t)$ is an extremely hard condition to satisfy. In the marginal case, it requires the model to perform one-step generation directly from a Gaussian distribution. This is a significantly harder problem—essentially what Consistency Models are designed to do.
> > >
> > > In fact, there is no quadratic runtime. We use $M \\ll N$ diffusion steps in the estimator, resulting in $O(MN)$. Our experiments show that $M=50$ for black hole is enough to achieve state of the art inference. While one-step generation is indeed hard, *few-step* models require less steps than diffusion models for the same quality (e.g. https://arxiv.org/pdf/2310.14189, Table 2), and could speed up our method by a factor when learned on $p(x \\mid x_t)$.
> > >
> > > > Langevin Dynamics [...] can also converge to the true posterior, but it avoids these expensive and impractical estimations entirely.
> > >
> > > Thanks for the suggestion, now we understand. We find that **we outperform Langevin Dynamics** as in https://arxiv.org/abs/1907.05600 on the Bayesian Inference tasks:
> > >
> > > | Method | Task 1 | Task 2 | Task 3 | Task 4 | Task 5 | Average |
> > > |--|--|--|--|--|--|--|
> > > | CBG-grad free (ours) | 0.505 (0.004) | 0.513 (0.009) | 0.584 (0.063) | **0.507 (0.006)** | **0.525 (0.028)** | **0.527** |
> > > | Langevin with REINFORCE | **0.502 (0.003)** | 0.514 (0.004) | 0.604 (0.009) | 0.705 (0.015) | 0.540 (0.013) | 0.573 |
> > > | Langevin with approximation | 0.506 (0.002) | 0.520 (0.005) | 0.633 (0.017) | 0.642 (0.083) | 0.538 (0.030) | 0.568 |
> > > | MCMC | 0.514 (0.005) | 0.511 (0.004) | 0.627 (0.005) | 0.512 (0.007) | 0.603 (0.046) | 0.553 |
> > >
> > > We test and tune these variants:
> > >
> > > - Langevin with REINFORCE uses our REINFORCE estimator from our CBG-grad free algorithm to estimate $\nabla_{x_t} \\log p(x_t \\mid y)$.
> > > - Langevin with approximation at every diffusion steps performs Langevin sampling on $p(x_t)p(y|x_0=x_t)$.
> > > - MCMC uses NUTS sampler (Hoﬀman & Gelman. JMLR 2014), suggestion by reviewer Qqm4
> > >
> > > All methods, including ours, use identical compute.
> > >
> > > We hope this addresses the remaining concerns.

---

### Decision · Program_Chairs · 2026-04-30

**Decision:**

Accept (regular)

**Comment:**

The paper investigates the issue of bias when using guidance to steer a pre-trained diffusion model toward a posterior distribution. Doing so requires gradients of log p(y | x_t) with respect to x_t. Correctly writing this gradient requires an integral over x. Existing approaches approximate the integral in various ways. The paper argues that existing approaches are biased because they do not give consistent estimators, and proposes two consistent gradient estimators that lead to correct posterior distributions. Experiments show that the corrected samplers outperform baselines on small Bayesian inference tasks and a black-hole imaging task.

Reviewers found the conceptual contribution (clearly pinpointing the bias in posterior approximations due to guidance approximations) strong and felt that the proposed solutions and empirical evaluation were sound. There were a few critiques: (1) increased computational cost, (2) lack of comparisons to methods like MCMC for the Bayesian inference tasks, which also converge to the correct posterior. The rebuttal added MCMC comparisons, discussed the intrinsic need for increased computation, and added a qualitative experiment on a larger image task. There was also a small discussion about prior work, which was mostly resolved.

Overall, the paper seems to make a useful conceptual contribution toward understanding the use of diffusion models with guidance to sample from Bayesian posterior distributions, and provides some promising empirical results on a limited set of problems. Its practical impact is not yet clear, but may warrant future study.